# WISE: Rethinking the Knowledge Memory for Lifelong Model Editing of Large Language Models

**Peng Wang**[1*]  **Zexi Li**[1*]  **Ningyu Zhang**[1†]  **Ziwen Xu**[1]  **Yunzhi Yao**[1]
**Yong Jiang**[2]  **Pengjun Xie**[2]  **Fei Huang**[2]  **Huajun Chen**[1,3†]
[1] Zhejiang University    [2] Alibaba Group
[3] Zhejiang Key Laboratory of Big Data Intelligent Computing
{peng2001,zexi.li,zhangningyu}@zju.edu.cn

## Abstract

Large language models (LLMs) need knowledge updates to meet the ever-growing world facts and correct the hallucinated responses, facilitating the methods of lifelong model editing. Where the updated knowledge resides in memories is a fundamental question for model editing. In this paper, we find that editing either long-term memory (direct model parameters) or working memory (non-parametric knowledge of neural network activations/representations by retrieval) will result in an impossible triangle—reliability, generalization, and locality can not be realized together in the lifelong editing settings. For long-term memory, directly editing the parameters will cause conflicts with irrelevant pretrained knowledge or previous edits (poor reliability and locality). For working memory, retrieval-based activations can hardly make the model understand the edits and generalize (poor generalization). Therefore, we propose WISE to bridge the gap between memories. In WISE, we design a dual parametric memory scheme, which consists of the main memory for the pretrained knowledge and a side memory for the edited knowledge. We only edit the knowledge in the side memory and train a router to decide which memory to go through when given a query. For continual editing, we devise a knowledge-sharding mechanism where different sets of edits reside in distinct subspaces of parameters and are subsequently merged into a shared memory without conflicts. Extensive experiments show that WISE can outperform previous model editing methods and overcome the impossible triangle under lifelong model editing of question answering, hallucination, and out-of-distribution settings across trending LLM architectures, e.g., GPT, LLaMA, and Mistral[‡].

## 1 Introduction

Large language models (LLMs) show emergent intelligence when scaling the number of parameters and data [1–4], which reveals the sparks of artificial general intelligence [5]. However, when deployed, LLMs still make mistakes [6], generating responses with hallucinations [7], bias [8], and factual decays [9]. On the other hand, the world's knowledge is ever-growing, so the up-to-date knowledge is usually different from the one during LLMs' pretraining [10]. Many such errors and emerging facts will arise sequentially in deployment, some of which have to be addressed timely and efficiently without waiting for retraining or finetuning [11, 12]. Also, retraining or finetuning is often too computationally expensive [13, 10], which is not sustainable for lifelong growing knowledge. Therefore, *lifelong model editing* [10] was proposed to remedy the continual knowledge updates and injections for LLMs in a cheap and timely manner.

---

[*]    Equal contribution.
[†]    Corresponding Author.
[‡] Code is available at https://github.com/zjunlp/EasyEdit.

38th Conference on Neural Information Processing Systems (NeurIPS 2024).

An effective lifelong model editing approach should satisfy the following properties [14, 15, 11, 16, 17]: **i) reliability**, the model can remember both current and previous edits after sequential editing; **ii) locality**, model editing will not influence inherent pretrained knowledge which is irrelevant to the edited knowledge; **iii) generalization**, the model is not just merely memorizing the query-target pairs; instead, it should understand and generalize when given other forms of queries with the same knowledge. We compare existing model editing and continual learning methods on the three metrics in Figure 1 and find that *it seems to be an impossible triangle—reliability, generalization, and locality* can not be realized at the same time in the continual editing settings. We find that where the updated knowledge resides in memories affects editing performances, and previous methods can be generally divided into editing either long-term memory, e.g., ROME [18], MEMIT [19], and FT-EWC (Finetuning with Elastic Weight Consolidation [20], a continual learning method), or working memory, e.g., GRACE [10]. Note that the categorization of long-term and working memories is derived from human recognition [21, 22] and neuroscience [23] which has recently been adopted in the study of LLMs [24–27]. Model editing of long-term memory refers to directly editing the model parameters, which contain generalizable parametric knowledge [28, 24]. However, editing long-term memory will cause conflicts with previous pretrained knowledge, resulting in poor locality (e.g., ROME and FT-EWC in Figure 1). Working memory refers to the non-parametric

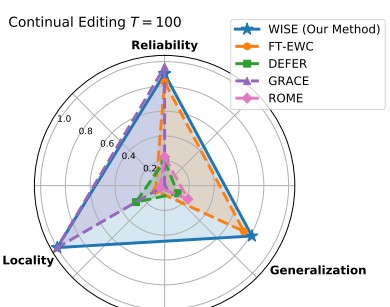

Figure 1: **Metric triangle among reliability, generalization, and locality.** ZsRE dataset, number of continual edits $T = 100$, LLaMA-2-7B. Editing methods based on long-term memory (ROME and FT-EWC) and working memory (DEFER and GRACE) show the impossible triangle in metrics, while our WISE is leading in all three metrics.

knowledge of neural network activations/representations by retrieval, and it does not change the network parameters [24]; instead, it replaces the representations by retrieval at working (inference) time, like GRACE. GRACE's working memory shows promising results in reliability and locality, but in our experiments, it shows poor generalization since retrieval-based representations can hardly make the model understand the edits and generalize to different queries. It reveals that long-term memory and working memory both have drawbacks for lifelong model editing, though there were some special memory designs for LLM architectures, like MemorryLLM [28], SPALM [27], and Memoria [25], they change the architectures and cannot be directly applied for different LLMs. Intuitively, there is a gap between editing working and long-term memories, thus, in this paper, we study:

> *What is the better memory mechanism for lifelong model editing to break the impossible triangle?*

Human brains contain the left and right hemispheres, which have different divisions as studied in recognition science [29, 30], e.g., the left brain is typically associated with logical tasks while the right brain is more involved in intuitive processes. This inspires us to design **WISE**, which makes model editor *WISER* in *memories*. WISE contains a dual parametric memory mechanism for LLMs' editing: the main memory for the pretrained knowledge and a side memory for the edited knowledge, realizing both long-term memory's generalization and retrieval-based working memory's reliability and locality. The side memory is a form of mid-term memory. We only edit the knowledge in the side memory and train a router to decide which memory to go through when given a query. For continual editing, we design a knowledge-sharding mechanism where different sets of edits reside in distinct and orthogonal subspaces of parameters. These are then merged into a common side memory without conflicts. Our contributions are as follows:

- We identify the pitfalls of current model editing methods in lifelong settings, that is, the impossible triangle among—reliability, generalization, and locality. Behind the impossible triangle, we find there is a gap between editing long-term memory and working memory.
- We propose WISE, with a side parametric memory as the mid-term memory, realizing the advantages of both parametric long-term memory and retrieval-based working memory. We design memory routing, sharding, and merging modules in WISE, making WISE lead in continual knowledge editing, reaching the three metrics better simultaneously.
- Extensive experiments on GPT, LLaMA, and Mistral across QA, Hallucination, and out-of-distribution datasets validate the effectiveness of WISE for lifelong model editing.

## 2 Methodology

### 2.1 Preliminaries: Lifelong Model Editing

We focus on lifelong model editing problem [10, 11], which can ensure hundreds or even thousands of sequential edits on LLMs to make the outputs of target queries align with human expectations while maintaining LLMs' previous knowledge and capability. Let $f_\Theta : \mathbb{X} \mapsto \mathbb{Y}$, parameterized by $\Theta$, denote a model function mapping an input $\mathbf{x}$ to the prediction $f_\Theta(\mathbf{x})$. The initial model before editing is $\Theta_0$, which is trained on a large corpus $\mathcal{D}_{\text{train}}$. When the LLM makes mistakes or requires injections of new knowledge, it needs model editing with a time-evolving editing dataset as $\mathcal{D}_{\text{edit}} = \{(\mathcal{X}_e, \mathcal{Y}_e) | (\mathbf{x}_1, \mathbf{y}_1), ..., (\mathbf{x}_T, \mathbf{y}_T)\}$. At the time step $T$, a model editor (ME) takes the $T$-th edit and the LLM of the $T-1$ time step $f_{\Theta_{T-1}}$ as inputs and produce the revised LLM model $f_{\Theta_T}$ following the equation below:

$$f_{\Theta_T} = \text{ME}(f_{\Theta_{T-1}}, \mathbf{x}_T, \mathbf{y}_T), \quad \text{s.t. } f_{\Theta_T}(\mathbf{x}) = \begin{cases} \mathbf{y}_e & \text{if } \mathbf{x} \in \mathcal{X}_e, \\ f_{\Theta_0}(\mathbf{x}) & \text{if } \mathbf{x} \notin \mathcal{X}_e. \end{cases} \quad (1)$$

Equation 1 describes that after model editing, the LLM should make the correct prediction on the current edit as $f_{\Theta_T}(\mathbf{x}_T) = \mathbf{y}_T$, while also preserving knowledge from past editing instances $(\mathbf{x}_{<T}, \mathbf{y}_{<T}) \in \mathcal{D}_{\text{edit}}$ as well as maintaining capability of $f_{\Theta_0}$ on the irrelevant data when $x \notin \mathcal{X}_e$, especially for general training corpus $\mathcal{D}_{\text{train}}$.

### 2.2 Rethinking the Memory Design of Lifelong Model Editing

Table 1: **Comparison of current model editing methods.** "✓" refers to "yes" and "well-supported", ✗ refers to "no" or "badly-supported", and "○" refers to "less-supported". The three metrics of Reliability, Generalization, and Locality denote the performances on lifelong (continual) editing.

| Methods | Long-term Memory | Working Memory | Parametric Knowledge | Retrieval Knowledge | Whether Lifelong | Reliability | Generalization | Locality |
|---|---|---|---|---|---|---|---|---|
| FT-EWC | ✓ | ✗ | ✓ | ✗ | ✓ | ✓ | ✓ | ✗ |
| ROME/MEMIT | ✓ | ✗ | ✓ | ✗ | ✗ | ✗ | ✗ | ✗ |
| MEND | ✓ | ✗ | ✓ | ✗ | ✗ | ✗ | ✗ | ✗ |
| SERAC/DEFER | ✗ | ✓ | ✓ | ✓ | ✓ | ○ | ✗ | ○ |
| GRACE | ✗ | ✓ | ✗ | ✓ | ✓ | ✓ | ✗ | ✓ |
| **WISE** | ✓ | ✓ | ✓ | ✓ | ✓ | ✓ | ✓ | ✓ |

In Table 1, we compare current model editing methods in terms of memory types and lifelong editing abilities. FT-EWC [20], ROME [18], MEMIT [19], and MEND [31] edit the long-term memory stored in the LLMs' model parameters, but they either do not support continual editing or have negative effects on irrelevant knowledge (poor locality). GRACE [10] is designed for lifelong editing via retrieval-based working memory. The retrieval codebook can avoid the conflicts of irrelevant knowledge, but GRACE fails to generalize due to its codebook being a non-parametric knowledge representation that solely memorizes queries without comprehension. It is worth noting that SERAC [32]/DEFER [10] uses working memory that is stored in additional small models: a scope classifier and a counterfactual model, whose knowledge is parametric. However, the small counterfactual model cannot match the expressiveness and generalization capabilities of LLM itself, making it challenging for the edited knowledge to generalize effectively.

To enable effective lifelong model editing, the method should take advantage of both LLM parameters' long-term memory and retrieval-based working memory. Therefore, we propose WISE as follows.

### 2.3 WISE: Side Memory with Knowledge Sharding, Merging, and Routing

As illustrated in Figure 2, WISE comprises two key components: 1) **Side Memory Design**: i) *side memory*: side memory is a memory container that is initialized as a copy of LLM's certain FFN layer, storing the stream of edits; ii) *memory routing mechanism*: similar to retrieval, a routing activation component is adopted to identify the scope of edits, routing the main (original) or side memories during inference; 2) **Knowledge Sharding and Merging**: i) *knowledge in random memory subspaces*: to make the edits in appropriate knowledge density and avoid forgetting, we shard the side memory into several random subspaces for editing; ii) *knowledge merging*: we leverage model merging techniques to merge different memory shards into one side memory without loss of knowledge.

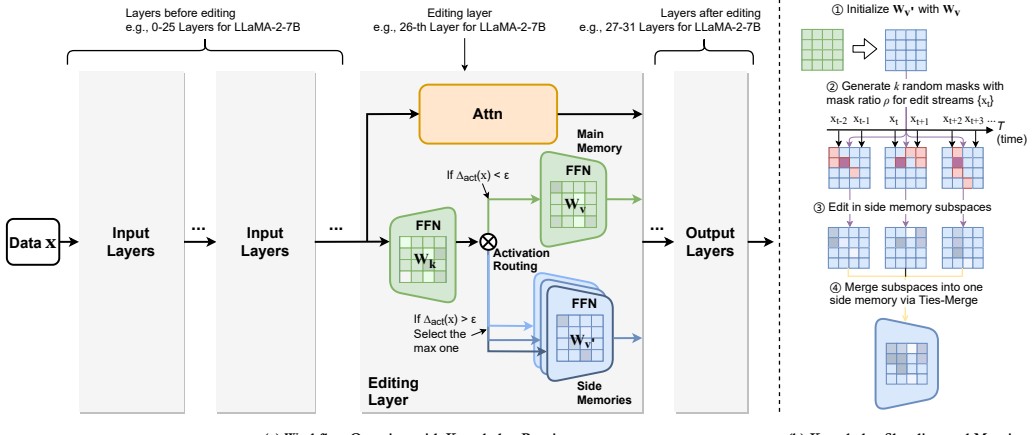

Figure 2: **Overview of WISE.** Side memory (in blue) and main memory (in green) store edited and pretrained knowledge, respectively. Note: during inference, if WISE-Retrieve, the activation routing will retrieve and select one side memory with maximal activation score.

### 2.3.1 Side Memory Design

**Side memory in FFN's value matrix.** Each layer in a Transformer contains a multi-head self-attention (MHA) mechanism and a feed-forward network (FFN), where the FFN constitutes two-thirds of the model parameters [33]. The question of how Transformers retrieve and utilize stored knowledge remains unresolved [18, 34], yet past works [31, 33] have demonstrated that editing the weights of the FFN is consistently more effective for LLMs. The FFN typically consists of key-value linear matrices: $\mathbf{W}_k, \mathbf{W}_v$, i.e., two multi-layer perceptron (MLP) layers. For the output of attention feature $\mathbf{f}$, the computation of the feed-forward network, omitting the bias terms, can be represented as:

$$\text{FFN}(\mathbf{f}) = \mathbf{a} \cdot \mathbf{W}_v = \sigma(\mathbf{f}^\top \cdot \mathbf{W}_k) \cdot \mathbf{W}_v, \tag{2}$$

where $\sigma$ is a nonlinear activation function (e.g. SwiGLU, GeLU), and $\mathbf{a}$ represents the activation values of the first MLP layer. Following previous works [18, 33], we edit the value matrix $\mathbf{W}_v$ of the chosen FFN layer.

However, directly editing the value matrix may cause forgetting and side effects in a lifelong setting. Thus, we **copy a value matrix as side memory and edit the side memory instead of the original matrix (main memory)**. Specifically, the side memory is initialized with the copy of main memory as $\mathbf{W}_{v'} \leftarrow \mathbf{W}_v$. Given the side memory, the new output is expressed as $\text{FFN}_s(\mathbf{f}) = \mathbf{a} \cdot \mathbf{W}_{v'}$. We will introduce how to update the side memory in Section 2.3.2.

**Locating side memory's FFN layer.** Transformer LLMs have been widely demonstrated to encode "lower-level" information (e.g., parts of speech) in earlier layers while processing more advanced linguistic phenomena like anaphora and coreference in later layers [35–37]. Representations in later hidden layers propagate through residual connections without drastic changes [38, 18], enabling effective early exit in LLMs [39, 40]. Therefore, to minimize the side effects of editing and adjust advanced linguistic phenomena, we target mid-to-late layers (e.g. 27) for side memory. Further analysis of layer selection is provided in Section 3.3.

**Routing between side memories and main memory.** Similar to the retrieval-based methods [10, 32], during inference, it is needed to decide whether the main memory or the side memory is used. If a given query is within the scope of previous edits, the side memory is used; otherwise, the main memory. Inspired by [11], we introduce a routing activation indicator, given an input $\mathbf{x}$, it is formulated:

$$\Delta_{\text{act}}(\mathbf{x}) = \|\mathcal{A}(\mathbf{x}) \cdot (\mathbf{W}_{v'} - \mathbf{W}_v)\|_2, \tag{3}$$

where $\mathcal{A}(\cdot) = \mathbf{a}$ is the activation of the side memory's corresponding FFN layer in Equation 2. We want the activation indicators of editing queries to be larger than the ones of irrelevant queries by a large margin, which is:

$$\min\{\Delta_{\text{act}}(\mathbf{x}_e)|\mathbf{x}_e \in \mathcal{D}_{\text{edit}}\} \gg \max\{\Delta_{\text{act}}(\mathbf{x}_i)|\mathbf{x}_i \in \mathcal{D}_{\text{irr}}\}, \tag{4}$$

where $\mathcal{D}_{\text{irr}}$ is the irrelevant dataset which includes $\mathcal{D}_{\text{train}}$.

To achieve the above objective, we design a margin-based loss function during editing training, similar to contrastive [41] or triplet loss [42]. The margin-based loss function for routing activation is:

$$L_a = \min_{\mathbf{W}_{v'}} \{\max(0, \Delta_{\text{act}}(\mathbf{x}_i) - \alpha) + \max(0, \beta - \Delta_{\text{act}}(\mathbf{x}_e)) + \max(0, \gamma - (\Delta_{\text{act}}(\mathbf{x}_e) - \Delta_{\text{act}}(\mathbf{x}_i)))\}, \quad (5)$$

$$\text{s.t. } \mathbf{x}_e \in \mathcal{D}_{\text{edit}}, \mathbf{x}_i \in \mathcal{D}_{\text{irr}}.$$

Equation 5 aims that for all queries of irrelevant examples $\mathbf{x}_i$, the activation indicators should be less than threshold $\alpha$, and for the edit samples $\mathbf{x}_e$, the activations should be larger than threshold $\beta$, with a certain distance $\gamma$ between $\Delta_{\text{act}}(\mathbf{x}_e)$ and $\Delta_{\text{act}}(\mathbf{x}_i)$.

In the continual stream of incoming edits, the smallest activation indicator within the edits is updated and saved: $\epsilon = \min\{\Delta_{\text{act}}(\mathbf{x}_e)|\mathbf{x}_e \in \mathcal{D}_{\text{edit}}\}$. We aim to recognize the local scope of edits in this form. During inference, if the activation indicator of a new input is greater than $\epsilon$, WISE will use the side memory $\mathbf{W}_{v'}$; otherwise, using the main memory $\mathbf{W}_v$. Thus, given the query $\mathbf{x}$, the output of the targeted FFN in Equation 2 is replaced by:

$$\text{FFN}_{\text{out}}(\mathbf{x}) = \begin{cases} \mathcal{A}(\mathbf{x}) \cdot \mathbf{W}_{v'} & \text{if } \|\mathcal{A}(\mathbf{x}) \cdot (\mathbf{W}_{v'} - \mathbf{W}_v)\|_2 > \epsilon, \\ \mathcal{A}(\mathbf{x}) \cdot \mathbf{W}_v & \text{otherwise.} \end{cases} \quad (6)$$

### 2.3.2 Knowledge Sharding and Merging

How to effectively and efficiently store continual knowledge in model parameters is important for lifelong editing. We introduce the notion of "*knowledge density*" (similar to knowledge capacity [43]) that describes how many pieces of knowledge are stored per parameter on average. There is an editing dilemma w.r.t. knowledge density: i) If only a few edits are made for full fine-tuning or editing the entire memory, the knowledge density is low, which may lead to overfitting. ii) If numerous edits are made within a common and limited parameter space, the knowledge density is high, resulting in conflicts within the edited knowledge and potentially causing catastrophic forgetting. To remedy this dilemma, we propose a knowledge sharding and merging mechanism to divide the edits into several shards, store them in different parameter subspaces, and merge them into a common side memory.

**Knowledge in random memory subspaces.** We edit the side memory $\mathbf{W}_{v'}$. We divide $n$ edits into $k$ shards, copy the side memory for $k$ times, and generate $k$ random gradient mask with mask ratio $\rho$ for each copy of side memory. A random gradient mask $\mathbf{M}_i \in \{0, 1\}^{|\mathbf{W}_{v'}|}, i \in [k]$ is a binary mask whose proportion of 1 is $\rho$ [44]. For edit shard $i, i \in [k]$, we edit the knowledge into the subspace $\mathbf{M}_i$ as follows:

$$\mathbf{W}_{v'}^i \leftarrow \mathbf{W}_{v'}^i - \eta(\mathbf{M}_i \odot \mathbf{g}_i(\mathbf{W}_{v'}^i)), \quad (7)$$

where $\mathbf{W}_{v'}^i$ is the $i$-th copy of the side memory, $\eta$ is the learning rate, $\mathbf{g}_i(\cdot)$ is the gradient of the $i$-th shard of edits, and the gradient is the autoregressive loss plus the routing activation loss $L_a$(Equation 5): $L_{\text{edit}} = -\log P_{W_{v'}}(\mathbf{y}_e|\mathbf{x}_e) + L_a$.

The random mask of gradients freezes the parameters intact when the elements are 0 and updates the weights when the elements are 1. It is superior to pruning because it does not harm the network performance while regularizing optimization in a subspace [44]. In addition, the $\rho$ subspace will have higher knowledge density when $k \cdot \rho < 1$, resulting in higher generalization (e.g., Figure 5). Also, different shards of edits have different random masks, and due to the (sub)orthogonality of random masks, different shards will not conflict with each other. Therefore, we can non-destructively merge the $k$ copies of side memory into one.

**Knowledge merging.** We merge the $k$ subspace pieces of side memory into one. Because we randomly generate the subspace masks, different random masks will have some overlapping elements and some disjoint elements, following the theorem below:

**Theorem 2.1 Subspace Overlap.** *Generate $k$ memory subspaces $\mathbf{W}_{v'}^i, i \in [k]$ by random mask with 1's ratio $\rho$, so each memory has $\rho \cdot |\mathbf{W}_{v'}|$ active trained parameters. For any two subspaces $\mathbf{W}_{v'}^i$ and $\mathbf{W}_{v'}^j$ $i \neq j; i, j \in [k]$, there are $\rho^2 \cdot |\mathbf{W}_{v'}|$ active parameters that are overlapped. For all $k$ subspaces, there are $\rho^k \cdot |\mathbf{W}_{v'}|$ overlapped active parameters.*

The theorem shows that larger $\rho$ will cause more overlap of subspace parameters, and the proof is in Appendix C. We find that this overlap is helpful in playing the role of "anchors" for knowledge merging (See Figure 5 and Appendix B.5). However, knowledge conflicts also exist in the overlapped parameters, so we leverage the recent task arithmetic model merging technique Ties-Merge [45] to

Table 2: **Main editing results for QA setting (ZsRE dataset).** $T$: Num Edits.

| Method | QA | | | | | | | | | | | | | | | |
|---|---|---|---|---|---|---|---|---|---|---|---|---|---|---|---|---|
| | $T=1$ | | | | $T=10$ | | | | $T=100$ | | | | $T=1000$ | | | |
| | Rel. | Gen. | Loc. | Avg. | Rel. | Gen. | Loc. | Avg. | Rel. | Gen. | Loc. | Avg. | Rel. | Gen. | Loc. | Avg. |
| LLaMA-2-7B | | | | | | | | | | | | | | | | |
| FT-L | 0.57 | 0.52 | 0.96 | 0.68 | 0.48 | 0.48 | 0.76 | 0.57 | 0.30 | 0.27 | 0.23 | 0.27 | 0.19 | 0.16 | 0.03 | 0.13 |
| FT-EWC | 0.96 | **0.95** | 0.02 | 0.64 | 0.82 | 0.76 | 0.01 | 0.53 | 0.83 | 0.74 | 0.08 | 0.55 | 0.76 | 0.69 | 0.08 | 0.51 |
| MEND | 0.95 | 0.93 | 0.98 | 0.95 | 0.26 | 0.28 | 0.28 | 0.27 | 0.00 | 0.00 | 0.00 | 0.00 | 0.00 | 0.00 | 0.00 | 0.00 |
| ROME | 0.85 | 0.80 | 0.99 | 0.88 | 0.64 | 0.62 | 0.75 | 0.67 | 0.23 | 0.22 | 0.04 | 0.16 | 0.01 | 0.01 | 0.00 | 0.01 |
| MEMIT | 0.84 | 0.81 | 0.99 | 0.88 | 0.58 | 0.58 | 0.85 | 0.67 | 0.02 | 0.02 | 0.02 | 0.02 | 0.04 | 0.04 | 0.02 | 0.03 |
| MEMIT-MASS | 0.84 | 0.81 | 0.99 | 0.88 | 0.75 | 0.72 | 0.97 | 0.81 | 0.76 | 0.68 | 0.85 | 0.76 | 0.69 | 0.65 | 0.62 | 0.65 |
| DEFER | 0.68 | 0.58 | 0.56 | 0.61 | 0.65 | 0.47 | 0.36 | 0.49 | 0.20 | 0.12 | 0.27 | 0.20 | 0.03 | 0.03 | 0.74 | 0.27 |
| GRACE | **0.99** | 0.36 | **1.00** | 0.78 | **0.96** | 0.16 | **1.00** | 0.71 | **0.96** | 0.15 | **1.00** | 0.70 | **0.93** | 0.08 | **1.00** | 0.67 |
| **WISE** | 0.98 | 0.92 | **1.00** | **0.97** | 0.94 | **0.88** | **1.00** | **0.94** | 0.90 | **0.81** | **1.00** | **0.90** | 0.77 | **0.72** | **1.00** | **0.83** |
| Mistral-7B | | | | | | | | | | | | | | | | |
| FT-L | 0.58 | 0.54 | 0.91 | 0.68 | 0.39 | 0.39 | 0.50 | 0.43 | 0.11 | 0.10 | 0.02 | 0.08 | 0.16 | 0.13 | 0.01 | 0.10 |
| FT-EWC | **1.00** | **0.99** | 0.01 | 0.67 | 0.84 | 0.78 | 0.02 | 0.55 | 0.82 | 0.72 | 0.09 | 0.54 | 0.76 | 0.69 | 0.09 | 0.51 |
| MEND | 0.94 | 0.93 | 0.98 | 0.95 | 0.01 | 0.01 | 0.02 | 0.01 | 0.00 | 0.00 | 0.00 | 0.00 | 0.00 | 0.00 | 0.00 | 0.00 |
| ROME | 0.79 | 0.77 | 0.98 | 0.85 | 0.58 | 0.57 | 0.75 | 0.63 | 0.05 | 0.05 | 0.02 | 0.04 | 0.04 | 0.04 | 0.02 | 0.03 |
| MEMIT | 0.81 | 0.79 | 0.99 | 0.86 | 0.46 | 0.45 | 0.61 | 0.51 | 0.00 | 0.00 | 0.01 | 0.00 | 0.04 | 0.04 | 0.02 | 0.03 |
| MEMIT-MASS | 0.81 | 0.79 | 0.99 | 0.86 | 0.74 | 0.71 | 0.97 | 0.81 | 0.73 | 0.71 | 0.88 | 0.77 | 0.73 | **0.70** | 0.62 | 0.68 |
| DEFER | 0.64 | 0.54 | 0.79 | 0.66 | 0.53 | 0.43 | 0.29 | 0.42 | 0.28 | 0.17 | 0.26 | 0.24 | 0.02 | 0.02 | 0.67 | 0.24 |
| GRACE | **1.00** | 0.36 | **1.00** | 0.79 | **1.00** | 0.15 | **1.00** | 0.72 | **1.00** | 0.15 | **1.00** | 0.72 | **1.00** | 0.02 | **1.00** | 0.67 |
| **WISE** | 0.98 | 0.97 | **1.00** | **0.98** | 0.92 | **0.89** | **1.00** | **0.94** | 0.87 | **0.80** | **1.00** | **0.89** | 0.70 | 0.67 | **1.00** | **0.79** |

relieve the conflicts. First, we compute the edit weight shift vectors $\mathrm{T}_e = \{\tau_e^i = \mathbf{W}_{v'}^i - \mathbf{W}_v | i \in [k]\}$. Then, we use Ties-Merge to merge the edit vectors into one:

$$\mathbf{W}_{v'} \leftarrow \mathbf{W}_v + \mathrm{Ties}(\mathrm{T}_e; \mathbf{W}_v). \tag{8}$$

Ties-Merge consists of three steps: i) trim: trim the redundant parameters for each task vector; ii) elect the sign: elect the signs of each parameter; ii) disjoint merge: compute the disjoint mean for each parameter which has the same and correct signs [45]. By Ties-Merge, different subspaces of knowledge are integrated into one with fewer conflicts. We study the effects of different merging techniques in Table 11 of Appendix B.2.

**Routing and retrieving among several side memories.** One single side memory has its limited knowledge capacity [43]. For the lifelong editing stream, we can produce several side memories and retrieve them via activation score routing. We compute different activation indicator scores of side memories and retrieve the top-1 during inference. This design is named WISE-Retrieve, which enables a more challenging lifelong editing scenario. For WISE with only one side memory, it is notated as WISE-Merge. For most of the experiments, we use WISE-Merge by default, and we compare WISE-Retrieve in Table 6 and Figure 6.

The pseudo-code of our method can be found in Algorithms 1 and 2.

## 3 Experiments

### 3.1 Experimental Settings and Evaluation Metrics

In the experiments, we compare the performance of different baselines and WISE in sequentially editing LLM models hundreds to thousands of times. In practice, we augment $\mathbf{x}_e$ by generating 10 random token sequences of length 10 using $f_\Theta$, enhancing editing generalization/adaptation to diverse contexts. We ensure that this augmentation with random tokens is applied across all baselines (See Appendix B.6, we ablate the contribution of Random Token).

**Datasets and Models.** We choose trending autoregressive LLM models **LLaMA-2-7B** [13], **Mistral-7B** [52], and **GPT-J-6B** [53, 54] for evaluation. The dataset details are in Table 3. Following [10], we evaluate WISE on the closed-book question-answering (QA) dataset **ZsRE** [46], and also evaluate its ability to correct **Hallucination** in SelfCheckGPT [48]. The

Table 3: Dataset statistics for main results. *Locality Data* is the irrelevant data of the editing process. $T$ is the number of samples. *Pre-edit* is the unedited model's performance on each dataset.

| SETTING | EDITING DATA | $T$ | Pre-edit (LLaMA/Mistral) | LOCALITY DATA |
|---|---|---|---|---|
| QA | ZsRE [46] | 1,000 | 0.36/0.39 ACC | NQ [47] |
| Halluc. | SelfCheckGPT [48] | 600 | 27.4/19.4 PPL | RedPajama [49] |
| OOD Gen. | Temporal [50] | 100 | 0.56 $\delta$-ACC (GPT-J) | Pile [51] |

**Temporal** dataset [50] is employed to test the out-of-distribution (OOD) generalization of editing. Since Temporal comprises emerging entities post-2019, we avoid using the latest LLMs in OOD experiments. Instead, we follow the original literature of the Temporal dataset [50] and adopt **GPT-J-6B** as the base model, which is pretrained on the Pile [51] with a cutoff in 2020. Implementation details and editing examples for each dataset and can be found in Appendix A.

Table 4: **Main editing results for Hallucination setting (SelfCheckGPT dataset).** $T$: Num Edits.

| | Hallucination | | | | | | | | | | | | | | | |
|---|---|---|---|---|---|---|---|---|---|---|---|---|---|---|---|---|
| | LLaMA-2-7B | | | | | | | | Mistral-7B | | | | | | | |
| | $T=1$ | | $T=10$ | | $T=100$ | | $T=600$ | | $T=1$ | | $T=10$ | | $T=100$ | | $T=600$ | |
| Method | Rel. (*PPL* ↓) | Loc. (↑) | Rel. (↓) | Loc. (↑) | Rel. (↓) | Loc. (↑) | Rel. (↓) | Loc. (↑) | Rel. (↓) | Loc. (↑) | Rel. (↓) | Loc. (↑) | Rel. (↓) | Loc. (↑) | Rel. (↓) | Loc. (↑) |
| FT-L | 4.41 | 0.96 | 12.57 | 0.71 | 33.06 | 0.41 | 69.22 | 0.26 | 25.03 | 0.38 | 100.00 | 0.03 | 1594.93 | 0.00 | - | - |
| FT-EWC | 2.56 | 0.24 | 3.63 | 0.09 | 2.10 | 0.16 | 4.56 | 0.24 | 1.75 | 0.04 | 3.05 | 0.09 | 4.73 | 0.17 | 5.46 | 0.25 |
| MEND | 5.65 | 0.87 | 11.01 | 0.86 | 10.04 | 0.88 | 1847.90 | 0.00 | 7.64 | 0.96 | 83.74 | 0.05 | 23114.94 | 0.01 | - | - |
| ROME | 1.68 | 0.99 | 2.04 | 0.94 | 94.15 | 0.05 | 104.93 | 0.02 | 2.04 | 0.99 | 3.45 | 0.92 | 103.75 | 0.03 | 241.17 | 0.01 |
| MEMIT | 1.66 | **1.00** | 2.36 | 0.97 | 76.65 | 0.05 | 107.61 | 0.02 | 1.64 | **1.00** | 15.89 | 0.89 | 97.23 | 0.04 | 132.30 | 0.02 |
| MEMIT-MASS | 1.66 | **1.00** | 1.61 | 0.99 | 7.18 | 0.96 | 13.47 | 0.94 | 1.64 | **1.00** | 2.78 | 0.99 | 3.22 | 0.97 | 7.28 | 0.95 |
| DEFER | **1.29** | 0.23 | 3.64 | 0.28 | 8.91 | 0.19 | 19.16 | 0.12 | 4.76 | 0.45 | 7.30 | 0.25 | 9.54 | 0.43 | 24.16 | 0.13 |
| GRACE | 2.21 | **1.00** | 8.67 | **1.00** | 9.67 | **1.00** | 9.34 | **1.00** | **1.39** | **1.00** | 5.97 | **1.00** | 9.53 | **1.00** | 9.57 | **1.00** |
| **WISE** | 1.91 | **1.00** | **1.04** | **1.00** | **1.14** | **1.00** | **3.12** | 0.99 | 1.40 | **1.00** | **2.56** | 0.94 | **1.31** | 0.99 | **5.21** | 0.93 |

**Baselines.** The baselines include methods of continual learning and model editing. We compare WISE against direct fine-tuning **FT-L** with an additional KL divergence loss [18], and continual learning fine-tuning based on Elastic Weight Consolidation (**FT-EWC**) [20]. We also compare WISE to other model editors, including 1) GPT-style editors based on causal tracing: **ROME** [18], **MEMIT** [19], and **MEMIT-MASS** (a batch-editing version of MEMIT); 2) hypernetwork-based editors: **MEND** [31]; and 3) the latest memory-based editors: **DEFER** (inspired by SERAC [32] for inference routing) and **GRACE** [10]. Details on all comparisons are found in Appendix A.2.

**Metrics.** Each edit example includes an edit descriptor (i.e., query) $\mathbf{x}_e$, its paraphrase prompts $\mathbf{x}_{e'}$ (if available) for testing generalization, and an unrelated statement $\mathbf{x}_{loc}$ for testing locality. For the editing dataset $\mathcal{D}_{\text{edit}} = \{(\mathcal{X}_e, \mathcal{Y}_e)\}$ with $T$ edits, we evaluate the final post-edit model $f_{\Theta_T}$ after the $T$-th edit example $(\mathbf{x}_T, \mathbf{y}_T)$. We evaluate the model editor's reliability and generalization using the metrics **Rel.** (a.k.a Edit Success Rate [10]) and **Gen.** (Generalization Success Rate [55]), while **Loc.** (Localization Success Rate [55]), defined as the post-edit model should not change the output of the irrelevant examples $\mathbf{x}_{loc}$, assesses specificity. We report these metrics and their mean scores, which are formally defined as:

$$\text{Rel.} = \frac{1}{T}\sum_{t=1}^{T}\mathbb{1}(f_{\Theta_T}(\mathbf{x}_e^t) = \mathbf{y}_e^t), \text{ Gen.} = \frac{1}{T}\sum_{t=1}^{T}\mathbb{1}(f_{\Theta_T}(\mathbf{x}_{e'}^t) = \mathbf{y}_e^t), \text{ Loc.} = \frac{1}{T}\sum_{t=1}^{T}\mathbb{1}(f_{\Theta_T}(\mathbf{x}_{loc}^t) = f_{\Theta_0}(\mathbf{x}_{loc}^t)), \quad (9)$$

where $\mathbb{1}(\cdot)$ is the indicator function. Notably, for the Hallucination dataset, following [10], we use the perplexity (PPL) to verify the locality, and there is no proper metric for generalization.

### 3.2 Main Results

**Competitive Performance of WISE.** The competitive performance of WISE is evident in Table 2 and 4, which compare its results with eight baselines on the QA (ZsRE) and Hallucination (SelfCheckGPT) settings. In general, we observe the followings: ❶ WISE outperforms existing methods on multiple tasks after long editing sequences; ❷ direct editing of long-term memory (ROME, MEMIT, etc.) creates conflicts with prior pretraining knowledge, resulting in poor locality; and ❸ retrieving working memory and modifying activations (GRACE, DEFER, etc) struggle to generalize to diverse queries.

In the **QA** setting, with $T = 1000$, WISE achieves average scores of 0.83 and 0.79 on LLaMA and Mistral, respectively, reflecting improvements of 18% and 11% over the nearest competitor. This demonstrates WISE's outstanding stability and effective management of long-sequential edits. While methods like MEND and ROME are competitive early in editing, they show clear shortcomings as the edit sequence extends. Directly editing long-term memory (e.g., MEMIT, FT-EWC, MEND) results in a significant

Table 5: **OOD results for Temporal dataset.** `GPT-J-6B` is used.

| | $T=10$ | | | | $T=75$ | | | |
|---|---|---|---|---|---|---|---|---|
| Method | Rel. | OOD Gen. | Loc. | Avg. | Rel. | OOD Gen. | Loc. | Avg. |
| *w/o Editing* | 0.56 | 0.21 | - | 0.39 | 0.56 | 0.21 | - | 0.39 |
| FT-EWC | 0.87 | 0.17 | 0.13 | 0.39 | 0.81 | 0.22 | 0.18 | 0.40 |
| ROME | 0.09 | 0.00 | 0.06 | 0.05 | 0.05 | 0.00 | 0.03 | 0.03 |
| MEMIT-MASS | 0.73 | 0.22 | 0.99 | 0.65 | 0.78 | 0.27 | 0.97 | 0.67 |
| DEFER | 0.68 | 0.33 | 0.08 | 0.36 | 0.52 | 0.26 | 0.08 | 0.29 |
| GRACE | 0.97 | 0.28 | **1.00** | 0.75 | **0.97** | 0.28 | **1.00** | 0.75 |
| **WISE** | **0.99** | **0.36** | 0.98 | **0.78** | 0.96 | **0.37** | **1.00** | **0.78** |

decline in Loc. When $T \in \{100, 1000\}$, this indicates that these methods cannot preserve LLMs' knowledge structure and significantly impair the model's generalization ability. GRACE excels in Loc. and Rel. (close to 1.00), however, it sacrifices generalization in continual editing. A possible reason is that token representation may not be suitable for measuring semantic similarity in autoregressive LMs, leading to paraphrase $\mathbf{x}_{e'}$ failing to achieve similarity matching with any CodeBook *Key* in GRACE (detailed in Appendix B.1). Overemphasis on preserving and precisely adapting training data (working memory) hampers adaptability to new contexts. In a nutshell, most previous methods struggle to balance Rel., Gen., and Loc., particularly in long-form editing tasks. In addition, the results of GPT-J-6B can be found in Figure 9 in the Appendix.

WISE also surpasses the baselines on the **Hallucination** dataset, maintaining the lowest perplexity scores of 3.12 and 5.21 at $T = 600$, with Loc. remaining above 0.93. We similarly observe

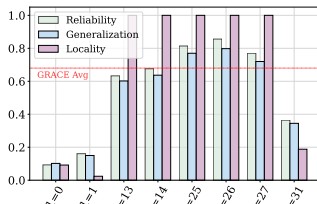



Figure 4: **Analysis of locating FFN layer of side memory for WISE.** ZsRE, LLaMA-2-7B.

Figure 5: **Analysis of different mask ratios $\rho$ and subspaces $k$ for WISE.** Left: Avg. performance of Rel., Gen., and Loc.; Right: the subspace overlap probability in Theorem 2.1. ZsRE, LLaMA-2-7B.

significant *PPL* increases for FT-L, MEND, and ROME in long-context editing tasks, while GRACE's performance is lackluster in LLM long texts (possibly due to the limited fitting capacity of the very small active trained parameters $|h^l|$ of GRACE).

**Out-of-Distribution Evaluation.** Ideally, model editing needs to generalize distributionally from formulaic editing examples to natural texts [50], where the distributional shift involves complexity rather than conventional domain shift [56]. Following [50], we evaluate the OOD generalization of editing methods on emerging entities using the temporal updating dataset, **Temporal**. Editing examples and evaluation metrics are provided in Appendix A.1. As shown in Table 5, WISE effectively handles out-of-distribution generalization tasks (achieving the best OOD Gen. and overall performance). DEFER delivers mediocre performance on OOD Gen. due to the limited capacity of the auxiliary model[14]. During the fine-tuning phase, GRACE and MEMIT focus on the representation $v*$ of a **single** input token after $\mathbf{W}_v$ (GRACE: last token, MEMIT: last subject token). However, regarding $v*$ the editing carrier encounters two problems: 1) the training objective is not aligned with the pretraining phase, and 2) the single representation limits the search scope of gradient descent, making it difficult to handle OOD generalization. WISE, on the other hand, avoids these challenges.

### 3.3 Further Analysis

**Visualization of WISE's Routing Activation.** To demonstrate the effectiveness of memory routing, we record the activation values $\Delta_{\text{act}}(\mathbf{x})$ of 1000 (QA, ZsRE)/600 (Halluc.) queries during the inference stage via knowledge merging into a single side memory. As shown in Figure 3, the purple horizontal line represents the activation threshold $\epsilon$ recorded during the editing phase. Almost all unrelated queries show low activations with values less than 10 in ZsRE and less than 20 in Halluc.; meanwhile, WISE accurately routes the editing prompt and unseen paraphrases into the side memory. This ensures editing locality and prevents excessive shifts from the pre-training distribution during lifelong editing.

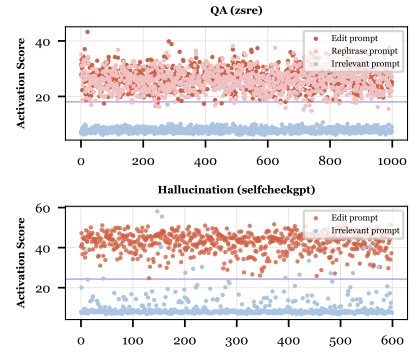

Figure 3: **Activations of the memory routing module of WISE when varying $T$.** X-axis: Num edits. LLaMA-7B.

**Localization Analysis of WISE's Side Memory.** To validate the benefits of editing mid-to-late layers, we select decoder layers from early, intermediate, mid-to-late, and late stages. As shown in Figure 4, the ablation results reveal that editing critical layers like the early and final layers (0, 1, 31) is ineffective, even resulting in a very low Loc. value of 0.096, which indicates a failure to recognize the editing scope. This may occur because the early layers represent fundamental grammatical information, and the final layer directly controls the decoding procedure, leading to poor editing of advanced language functions. Editing in the intermediate layers is suboptimal but still shows a markable improvement compared to early layers, possibly because intermediate layers start to integrate basic grammatical information with more complex semantic data. Notably, the mid-to-late layers demonstrate exceptional editing performance; for instance, selecting layer 26 results in an 80% success rate and generalization while maintaining 100% locality. This empirically supports our claim in Section 2.3.1 that the redundant mid-to-late layers [39] are ideal side memory layers and confirms the hierarchical nature of information processing in Transformer LLMs [57, 58].

**Analysis of $\rho$ and $k$ for WISE.** We analyze the important hyperparameters of WISE: the mask ratio $\rho$ and the number of subspaces $k$ in Figure 5. On the left figure, for $k = 2$, the best $\rho$ is 0.2,

satisfying $k * \rho = 0.4 < 1$, which implies the effectiveness of our subspace design that higher knowledge density will cause better generalization. When scaling $k$, we observe an increasing demand of $\rho$. From Theorem 2.1, the probability of subspace overlap is $\rho^k$, and we hypothesize that this overlap is important as an anchor for model merging. Interestingly, from the right figure, it can be observed that the optimal cases always have the $\rho^k$ closest to 0.03. This shows an inherent tradeoff between merge anchor and merge conflicts, and the subspace overlaps around 0.03 are optimal for the best performances. Such experiments indicate that 20% FFN parameters can accommodate at least 500 edited samples. When "mask memory exhaustion" occurs, we can allocate new mask parameters to store new knowledge. Using retrieve when knowledge isn't full and merging as needed to save memory, achieves true lifelong model editing.

**Scale Up to 3K of Edits.** We scale the number of continual edits to 3K in Table 6. We compare WISE-Merge, keeping one side memory by multi-time merging, and WISE-Retrieve, keeping several side memories by routing and retrieving among different side memories. For WISE-Retrieve, we show an upper bound "*oracle*", which always identifies the correct routing path. We observe that the WISE series maintains high scalability, consistently outperforming the strongest baselines including MEMIT-MASS and GRACE. WISE-Retrieve based on top-1 activation retrieval demonstrates the best results in 3K edits, showing the effectiveness of well-organized memory subspaces and routing strategies during editing. We note that the "*oracle*" exhibits marginal performance decline when scaling the edits from 2K to 3K, yet it demonstrates remarkable performance across all metrics. This underscores the potential of WISE to handle extremely long continual edits, contingent upon substantial improvement in the retrieval of side memories. Additionally, an appropriate replay of edits can further improve retrieval accuracy, as detailed in Appendix B.3.

Table 6: **Scaling to 3K edits of ZsRE.** `LLaMA-2-7B`.

| Method | $T = 2000$ | | | | $T = 3000$ | | | |
|---|---|---|---|---|---|---|---|---|
| | Rel. | Gen. | Loc. | Avg. | Rel. | Gen. | Loc. | Avg. |
| GRACE | **0.96** | 0.03 | 1.00 | 0.66 | **0.96** | 0.03 | 1.00 | 0.66 |
| MEMIT-MASS | 0.64 | 0.58 | 0.55 | 0.59 | 0.58 | 0.53 | 0.47 | 0.53 |
| WISE-Merge | 0.66 | 0.63 | 1.00 | 0.76 | 0.58 | 0.56 | 1.00 | 0.71 |
| WISE-Retrieve | 0.68 | **0.64** | **1.00** | **0.77** | 0.61 | **0.58** | **1.00** | **0.73** |
| WISE-Retrieve$_{oracle}$ | 0.77 | 0.72 | 1.00 | 0.83 | 0.75 | 0.70 | 1.00 | 0.82 |

**Contribution of Router designs in WISE.** Without the router strategy, all inputs either pass solely through the main or side memory. To further validate its effectiveness, we conduct additional ablations with $L_a$. WISE's performance on ZsRE is shown in Table 7. We observe the expected decrease in Loc. w.o. $L_a$, such as dropping from 1.00 to 0.72 at T=1000, reveals the router's effectiveness in identifying editing scopes, minimizing side effects, and retaining a substantial amount of pre-training knowledge.

Table 7: **Ablation study of Router (compared with Table 2).** `LlaMA`.

| WISE$_{w.o. L_a}$ | Rel. | Gen. | Loc. | Avg. |
|---|---|---|---|---|
| $T = 1$ | 1.00 | 0.96 | 0.93 -0.07 | 0.96 -0.01 |
| $T = 10$ | 0.93 | 0.90 | 0.88 -0.12 | 0.90 -0.04 |
| $T = 100$ | 0.92 | 0.85 | 0.81 -0.19 | 0.86 -0.04 |
| $T = 1000$ | 0.84 | 0.79 | 0.72 -0.28 | 0.78 -0.05 |

**Inference Time Analysis of WISE.** Figure 6 shows the inference time of a single instance for LLaMA after $t \in [0, 3000]$ editing steps, measured across 10 trials of each setting. Consistent with our expectations, we find that WISE-Merge incurs a constant inference delay (about 3%) as the editing stream expands. WISE-Retrieve, due to the introduction of retrieval routing, shows an increase in inference time as the number of edits increases, with a time cost increment of about 7% after 3K edits. Knowledge merging ensures that WISE-Merge only brings constant additional costs (0.64% extra parameters and 4% extra GPU VRAM, as detailed in Appendix B.7), contrasting with past memory-based works that continuously demand more available memory [10, 32].

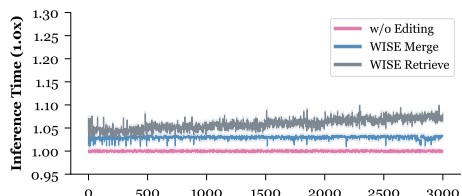

Figure 6: **Inference time of WISE when varying $T$.** ZsRE, `LLaMA-2-7B`.

## 4 Related Works

**Memory and Knowledge Injection of LLMs.** LLMs have long-term (episodic) and working memory [24, 25, 27]. Long-term memory is stored in model parameters, updatable via (re)pretraining [53], finetuning [59], and model editing [14]. Working memory resides in neuron activations, utilized during inference [24]. In-context learning and retrieval-based editing methods like GRACE contribute to working memory [60, 10]. However, whether finetuning or retrieval is debated [61, 62]. Also, current knowledge injection methods often suffer from computational overhead [13, 10], catastrophic forgetting [63], and overfitting [64]. Methods like MemorryLLM [28], SPALM [27], NKB [65], and Memoria [25] are proposed to improve the memories from the architecture design perspective.

**Model Editing of LLMs.** Model editing encompasses constrained finetuning, locating-and-editing, meta-learning, and retrieval-based methods. ROME identifies factual associations and edits efficiently using MLP-based memories [18], extended by MEMIT for mass-editing [19]. T-Patcher adds neurons for edits in LLMs' feed-forward layers [11]. Meta-learning methods like MEND decouple finetuning gradients to generalize edits [31], complemented by MALMEN addressing cancellation effects [15]. Retrieval-based methods like SERAC and GRACE improve working memory for editing [32, 10]. From single to mass editing and static to lifelong editing, model editing evolves to meet realistic demands. The latest efforts in lifelong editing such as LTE [66], MALMEN [15], and RECIPE [67] require extensive training with domain-specific edits before specific editing, yet we cannot predict the domain of upcoming edits in the editing flow and accessing these data is often impractical or unrealistic. It potentially increases the risks associated with retraining.

**Model Merging** Model merging [68], also known as model fusion [69, 70], studies how to aggregate different models' knowledge into one by parameter merging. However, in the research of linear mode connectivity, it is found that different minima of neural networks can hardly be merged into a generalized one even if trained on the same datasets from the same initialization (but with different random seeds) [71, 72]. The main reason is considered to be the permutation invariance property of deep neural networks, which means that the positions of neurons can be permuted without affecting the network function [71]; as a result, different minima reside in different loss basins [72]. To improve linear mode connectivity and model merging, methods like optimal transport [70, 73], re-basin [72], and training-time alignment [44] are developed. For the applications, model merging techniques can help to improve the generalization of federated learning [74, 75] and enable knowledge aggregation of different-task models in a task arithmetic way [76, 77]. Recently, methods like task arithmetic in tangent space [77], TIES-Merging [45], ZipIt! [78], and ColD fusion [79] have been proposed for deep model fusion of pretrained foundation models, such as CLIP, ViT, and large language models. Specifically, TIES-Merging [45] consists of trim, elect sign & merge pipeline, which inspires the merge process of side memories in our paper.

For detailed related works, please refer to Appendix D.

## 5 Limitations and Broader Impacts

Although WISE shows promising results in lifelong editing, it also has some limitations. One limitation is addressed in Table 6 that the side memory retrieval has room for improvement to reach the oracle. Also, in Figure 6, the inference time of WISE-Retrieve increases with ever-growing editing streams. However, the current limitations cannot outweigh the merits of WISE in that it currently reaches better performance in general for lifelong model editing. We bridge the gap between long-term and working memory, it may inspire further work on memory design for model editing or even LLM architecture. However, the application of such technologies should be guided by ethical considerations. Malicious users may attempt to edit LLMs to propagate hate, highlighting the need for safeguards to prevent abuse and mitigate harmful outcomes. Some current model editors update the model's weights directly, making edits hard to trace and withdraw. WISE uses a modular and non-destructive side memory, allowing users to discard it if edits are unnecessary or harmful, without modifications to the main LLMs.

## 6 Conclusion

In this paper, we point out the impossible triangle of current lifelong modeling editing approaches that reliability, generalization, and locality can hardly be achieved simultaneously. We find the reason behind this is the gap between working and long-term memory. Therefore, we propose WISE, consisting of side memory and model merging, to remedy the gap.

## Acknowledgements

We would like to express gratitude to the anonymous reviewers for their kind comments. This work was supported by the National Natural Science Foundation of China (No. 62206246, No. NSFCU23B2055, No. NSFCU19B2027), the Fundamental Research Funds for the Central Universities (226-2023-00138), Zhejiang Provincial Natural Science Foundation of China (No. LGG22F030011), Yongjiang Talent Introduction Programme (2021A-156-G), SMP-Zhipu.AI Large Model Cross-Disciplinary Fund, Ningbo Science and Technology Special Projects under Grant No. 2023Z212, Information Technology Center and State Key Lab of CAD&CG, Zhejiang University. We gratefully acknowledge the support of Zhejiang University Education Foundation Qizhen Scholar Foundation.

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

# Appendix

In the Appendix, we introduce more details along with additional experimental results, discussions, and related works:

- Appendix A: Experimental setups (cf. Section 3).
- Appendix B: More experimental results (cf. Section 2 and 3).
- Appendix C: Proof of the Theorem 2.1 (cf. Section 2).
- Appendix D: Additional discussions and more related works (cf. Section 4).

## A Implementation Details

### A.1 Description of Datasets

Table 8: Bolded text refers to the edit labels $\mathbf{y}_e$. Locality example $\mathbf{x}_{\text{loc}}$ is an unrelated query.

(a) **ZsRE, question-answering** editing dataset example.

| | |
|---|---|
| $\mathbf{x}_e, \mathbf{y}_e$ | Which continent is Berkner Island in? **South America** |
| $\mathbf{x}_{\text{loc}}$ | who gets the golden boot if its a tie? **shared** |
| $\mathbf{x}'_e, \mathbf{y}_e$ | On which continent is Berkner Island located? **South America** |

(b) **Hallucination** editing dataset example. In the original data [10], there is no paraphrase $x_{e'}$ so the measurement of Gen. metric is ignored here.

| | |
|---|---|
| $\mathbf{x}_e, \mathbf{y}_e$ | This is a Wikipedia passage about heinz christian pander. Heinz Christian Pander (1794 - 1865) was a German anatomist and embryologist who was born in Riga, Latvia. He studied medicine at the University of Dorpat and later at the University of Berlin. **In 1820, he took part in a scientific expedition to Bokhara as a naturalist.** |
| $\mathbf{x}_{\text{loc}}$ | Tired and restlessly, drifting in and out of sleep. Hearing crashing and banging, thinking the roof will cave in. Not alert enough to quite know what it was, I yelled loudly for whoever was making those noises at such an hour to stop. They heard and listened, I'm guessing |

**ZsRE** The ZsRE question-answering task [46] is extensively studied within the model editing literature [18, 19, 31, 15, 11], where each record contains an editing statement $\mathbf{x}_e$, a paraphrase prompt $\mathbf{x}'_e$, and a locality prompt $\mathbf{x}_{\text{loc}}$. We use the same train/test split as [31] (163196/19086). Notably, only MEND requires fitting a hypernetwork on the training set; other methods discard the training set and perform edits and evaluations on the test set. In practice, we randomly sample 1K and 3K records from the test set to form the edit sets in Section 3.2 and 3.3.

**Hallucination** We utilize the same dataset as GRACE, SelfCheckGPT [48], to assess the ability of Model Editors to mitigate hallucinations in autoregressive LMs. This setting involves editing highly inaccurate sentences (sourced from GPT-3 [80]) and replacing them with corresponding sentences from actual Wikipedia entries. This dataset aligns more closely with real-world deployment scenarios where models trigger "unexpected behaviors," and the token length of edits is significantly longer than in past datasets, making it a more challenging editing setting. Unlike GRACE, which used GPT2-XL (1.5B) [81], our main experiments deploy larger LLMs, LLaMA and Mistral, both with 7B parameters, we measure retention of pretraining data ($\mathbf{x}_{\text{loc}}$) from the base model: Red-Pajama [49], a public version of LLaMA's pretraining data. Some of the exceptionally long editing samples cannot even be accommodated on an NVIDIA A800 (80GB) due to resource limitations. As shown in Figure 7, the original dataset provided by GRACE, after tokenization with LLAMATO-KENIZER, has length distributions ranging from [17,390]. The dimension of a single MLP layer in `llama-2-7b-hf` is (11008, 4096) [§]. Theoretically, fine-tuning an input of length 390 with default

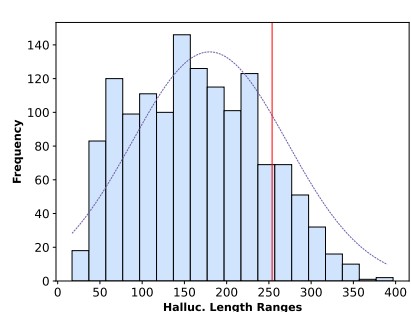

Figure 7: Hallucination length statistics.

---

[§] https://huggingface.co/meta-llama/Llama-2-7b-hf

Table 9: **Temporal** OOD dataset example. Bolded text refers to the edit labels $\mathbf{y}_e$ and $\mathbf{y}_{\text{ood}}$.

| | |
|---|---|
| $\mathbf{x}_e, \mathbf{y}_e$ | Self-driving cars, **also known as autonomous vehicles, are vehicles that are capable of navigating and operating without human intervention. These innovative vehicles rely on a combination of advanced sensors, artificial intelligence, and computer algorithms to interpret their environment and make real-time decisions. With the potential to significantly impact numerous industries and sectors, self-driving cars have the ability to revolutionize transportation by enhancing safety, improving traffic flow, and increasing energy efficiency. However, challenges related to regulatory frameworks, ethical considerations, and public acceptance still need to be addressed before widespread adoption becomes a reality.** |
| $\mathbf{x}_{\text{loc}}$ | Apple has a new peach with the release of its 3.0GHz, 8-core Intel Xeon-based Mac Pro. The 8-core Mac Pro is powered bu two quad-core Intel Xeon C̈lov ertownp̈rocessors running at 3.0GHz. Apple also released a quad-core Mac Pro featuring two Dual-Core Intel Xeon Ẅoodcrestp̈rocessors. |
| $\mathbf{x}_e, \mathbf{y}_{\text{ood}}$ | Self-driving cars, **also known as autonomous cars or driverless cars, are vehicles capable of traveling without human input. These cars utilize a range of sensors, including optical and thermographic cameras, radar, lidar, ultrasound/sonar, GPS, odometry, and inertial measurement units, to perceive their surroundings. By interpreting sensory information, control systems in the car are able to create a three-dimensional model of its environment. Using this model, the car can then identify the best navigation path and develop strategies for managing traffic controls and obstacles. As self-driving car technology continues to advance, it is expected to have a significant impact on various fields such as the automotive industry, health, welfare, urban planning, traffic, insurance, and the labor market. The regulation of autonomous vehicles is also becoming an increasingly important topic of discussion.** |

full precision and the Adam optimizer would require (390+4+4+4) * (11008 * 4096 * 4) + 4 * 7B = **100.36GB** of VRAM (for activations, gradients, first-order, and second-order optimizers), exceeding the memory capacity of the NVIDIA A800. Consequently, we excluded excessively long samples (limiting tokenized lengths to 254) and ultimately retained 906 editing instances (compared to 1392 in GRACE). To facilitate a fair comparison with MEND, we specifically allocated a training set for MEND, with a final train/test split of 306/600. All methods were edited and evaluated on the test set.

**Temporal** [50] sources the prefix $\mathbf{x}_e$ from the first paragraph of an entity's Wikipedia page and samples a paragraph $\mathbf{y}_e$ discussed by GPT-4 [82] about the emerging entity $\mathbf{x}_e$, which is usually noisy but may contain helpful information. These are presented as editing prompts to Model Editors. For out-of-distribution (OOD) generalization to complex natural contexts (not fitted), $\mathbf{y}_{\text{ood}}$ is taken from the actual Wikipedia suffix of $\mathbf{x}_e$. This setup is utilized to evaluate the OOD generalization of Model Editors centered around a single canonical example. Consistent with previous work [10], the out-of-scope data $\mathbf{x}_{\text{loc}}$ is derived from the Pile [51], the pretraining corpus of GPT-J-6B. Examples from the dataset can be seen in Table 9. To measure the OOD generalization of editing methods for emerging entities, we perform model editing using standardized simple examples and then evaluate this behavior on more complex instances. Following [50], in a natural setting, no single correct continuation exists. Thus, we also use probability threshold-based evaluations, such as 80%, where the editing success rate evaluates whether the loss $L_{\mathbf{x}_e, \mathbf{y}_{\text{ood}}}$ for an example falls below $\delta = -\log(0.8)$, as indicated in the formula below. The intuition behind this is that many other plausible alternative continuations may exist.

$$\text{OOD Gen.} = \frac{1}{T} \sum_{t=1}^{T} \mathbb{1}\{(L_{\Theta_T}(\mathbf{x}_e, \mathbf{y}_{\text{ood}}) < \delta)\}. \tag{10}$$

### A.2 Descriptions of Compared Model Editors

**FT-L.** All other layers of the LLMs remain frozen, and only a single MLP layer is fine-tuned through autoregressive loss [18]. Additionally, we impose an $L_\infty$ norm constraint to prevent the parameters from deviating too far from the pretrained distribution.

**FT-EWC.** Elastic Weight Consolidation (EWC) has been demonstrated to mitigate catastrophic forgetting by updating weights using a Fisher information matrix, which is computed from past edits,

multiplied by a scaling factor $\lambda$ [20]. Following [10], we omit the constraints of the $L_\infty$ norm in this implementation.

**MEND.** MEND [31] transforms the gradients obtained from standard fine-tuning using a hyper-network that converts gradients decomposed into low rank (rank=1) into new gradients, which are then applied to the target layer for parameter updates. During the training phase, a small auxiliary hypernetwork receives editing examples $(\mathbf{x}_e, \mathbf{y}_e)$, and $\mathbf{x}_{loc}$. MEND's training loss comprises the standard autoregressive loss combined with the KL divergence loss of the model's output on $\mathbf{x}_{loc}$ before and after editing. This hypernetwork plays a crucial role during the editing procedure.

**ROME.** ROME [18] uses causal analysis to pinpoint knowledge within specific MLP layers and modifies the entire matrix through least squares approximation. It operates under the strong assumption that the MLP is the primary module for storing knowledge [33], and it injects a single piece of knowledge into the MLP at each iteration using a Lagrangian remainder.

**MEMIT.** Similarly, based on the assumption that the FFN serves as a knowledge key-value store, MEMIT [19] manipulates parameters of specific layers directly through least squares approximation. Unlike ROME, which updates a single layer, MEMIT is a multi-layer updating algorithm that supports simultaneous updates of hundreds or thousands of facts. For sequential model editing tasks, MEMIT requires immediate on-the-fly repairs when the model makes errors, expressed as $f_{\Theta_T} = \text{MEMIT}(f_{\Theta_{T-1}}, \mathbf{x}_T, \mathbf{y}_T)$, involving multiple operations on the original model.

**MEMIT-MASS.** Unlike sequential editing, MEMIT supports modification of multiple knowledge fragments in a batch mode, named **MEMIT-MASS**. Suppose we collect streaming errors as $(\mathcal{X}, \mathcal{Y}) = \{(\mathbf{x}_0, \mathbf{y}_0), (\mathbf{x}_1, \mathbf{y}_1), ..., (\mathbf{x}_T, \mathbf{y}_T)\}$ and inject them collectively into the MLP, it only involves a single editing operation on the original model as $f_{\Theta_T} = \text{MEMIT}(f_{\Theta_0}, \mathcal{X}, \mathcal{Y})$. Although this approach **loses the capability for on-the-fly repairs**, we still include this baseline in our experiments.

**DEFER.** In GRACE, a reimplementation of SERAC [32] is utilized, denoted as DEFER. For new inputs, DEFER includes a network $g$ (corresponding to the *scope classifier* in SERAC) that predicts whether to: 1) trust the prediction of the LLMs, or 2) trust the prediction of the new model. Here, the new model is configured as a single-layer linear network $o$ with a sigmoid activation function, corresponding to the *counterfactual model* in SERAC. During the editing process, $g$ and $o$ are fine-tuned jointly.

**GRACE.** GRACE [10] utilizes a discrete KEY-VALUE codebook and maintains the codebook throughout the editing flow by adding, expanding, and splitting KEYs. During the inference phase, it retrieves the nearest KEY and determines whether to replace the activation of the hidden layer output.

### A.3 Training Details and Hyperparameters

Except for MEMIT-MASS, the batch size for all methods is consistently 1 in sequential editing scenarios. All experiments are conducted using 3 NVIDIA A800 GPUs, with all tasks reproducible on a single A800. Editing ZsRE takes approximately 4 hours, while Hallucination requires around 6 hours. To ensure fair comparisons, unless otherwise specified (for some methods like MEND, ROME, and MEMIT, we follow the original literature by selecting the last few layers or using causal analysis to identify the target layers), the default target layers for editing on `LLaMA`, `Mistral`, and `GPT-J` are `model.layers[27].mlp.down_proj.weight`, `model.layers[27].mlp.down_proj.weight`, and `transformer.h[21].mlp.c_fc`, respectively.

For FT-L, we utilize a reimplementation from ROME [¶], employing the Adam [83] optimizer with consideration of learning rates at 1e-5, 1e-4, and 5e-4, and conducting gradient descents for 50 iterations, ultimately reporting the best results at a learning rate of 5e-4.

For FT-EWC, we follow the reimplementation in GRACE and its default settings, setting the learning rate at 1e-2, the $\lambda_{ewc}$ penalty factor at 0.1, and the number of replay instances at 10.

For the training phase of MEND, we adhere to the original paper, setting the learning rate at 1e-4, iterating 100K times, and employing early stopping at 30K, ultimately achieving an accuracy of 0.95 on the training set. Notably, we target the last few MLP layers as per the original literature, such as `model.layers[i].mlp.down_proj.weight`, `model.layers[i].mlp.gate_proj.weight`, `model.layers[i].mlp.up_proj.weight` in LLaMA, where $i \in [29, 30, 31]$.

For ROME and MEMIT, we follow the original literature on GPT-J using the default configurations, specifically the fifth layer and layers [3,4,5,6,7,8]. In LLaMA and Mistral, additional causal analysis is conducted to pinpoint the layers storing knowledge. As shown in Figure 8, an increasing trend in

---

[¶] https://github.com/kmeng01/rome

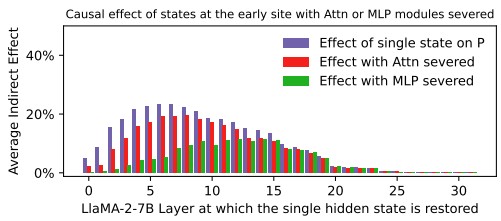
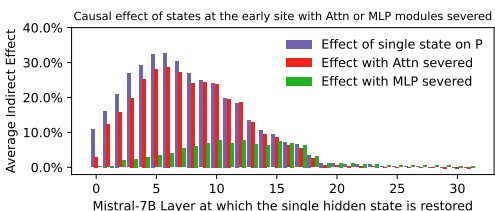

Figure 8: Mid-layer MLPs play a crucial mediating role in `LLaMA-2-7B` and `Mistral-7B`.

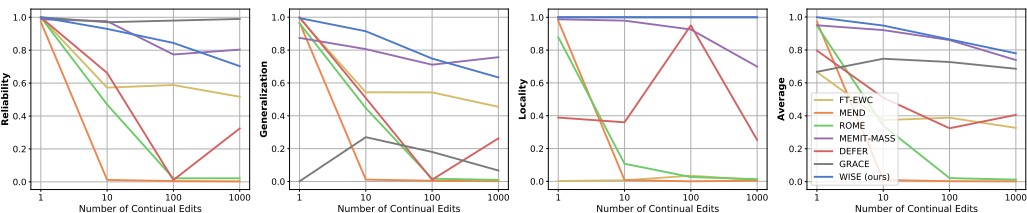

Figure 9: GPT-J-6B, ZsRE, continual editing.

the Average Indirect Effect of the MLP is observed across layers [4,5,6,7,8], suggesting that the model recalls factual knowledge here and passes the matured token distribution via residual connections to the last MLP. Thus, in LLaMA and Mistral, ROME edits the fifth layer, while MEMIT edits layers [4,5,6,7,8].

For DEFER, the original literature uses a learning rate of 1.0; however, we found it unfit for LLaMA and Mistral, with severe fluctuations in model loss. Therefore, we experiment with learning rates of 7e-5, 7e-4, and 1e-3, and ultimately report using 7e-5 (optimal).

For GRACE, we strictly follow the original literature, setting the learning rate at 1.0, and using `replace_last` to only replace the activation of the last token in autoregressive scenarios. After observing failures in generalization, we adjust various $\epsilon_{\mathrm{init}}$ values and discuss this more in Appendix B.1.

For WISE, the hyperparameters for the QA and Hallucination tasks are identical. We find that a learning rate of 1.0 with the SGD [84] optimizer is a good approach for stable training. The hyperparameters designed in the knowledge editing phase include the random masking probability $\rho$ and the routing threshold guidance $\alpha, \beta, \gamma$. In the knowledge merging phase, hyperparameters include the number of merges $k$ and the merging weights $\lambda$ for each MLP (we discuss the impact of $\rho$ and $k$ in Section 3.3). Theoretically, as the importance of knowledge in any MLP is considerable, we always average with $\lambda = 1/k$ across all experiments. These are shown in Table 10.

Table 10: WISE hyper-parameters during editing and merging.

| Hyper-Parameters | Values |
|---|---|
| Optimizer | SGD |
| LR $\eta$ | 1.0 |
| Mask Ratio $\rho$ | 0.2 |
| $\alpha$ | 5.0 |
| $\beta$ | 20.0 |
| $\gamma$ | 10.0 |
| Merge Weights $\lambda$ | 0.5 |
| Knowledge shards $k$ | 2 |

### A.4 Pseudo Code of WISE

The pseudo-code of the WISE editing stage is in Algorithm 1, and the one of the WISE inference stage is Algorithm 2.

## B More Experimental Results and Analyses

### B.1 On the Pitfall of GRACE: Generalization Collapses in Decoder-only LLMs

Here, we discuss why GRACE exhibits poor generalization when editing decoder-only LMs.

As shown in Figure 10, we continuously edit 15 samples $(\mathbf{x}_e, \mathbf{y}_e)$ using GRACE and observe the nearest codebook *Key* for their paraphrases $\mathbf{x}_{e'}$ and unrelated queries $\mathbf{x}_{\mathrm{loc}}$, as well as the governed *Deferral radii* $\epsilon$ of those *Keys*. When overlapping *Keys* exist, GRACE reduces the *Deferral radii* to split this *Keys* and then adds a new codebook entry, resulting in exponentially decaying of radii $\epsilon$ during the editing process. Though $\epsilon$ is initialized from a high $\epsilon_{\mathrm{init}}$, it will be small and ineffective after continuous edits. From Figure 10, we observe that GRACE is more likely to have a conservative

---

**Algorithm 1:** WISE Editing Stage

---

**Input**: The initial LLM model $f_{\Theta_0}$, the targeted FFN layer, the edit dataset $\mathcal{D}_{\text{edit}}$ whose length is $T$, the irrelevant dataset $\mathcal{D}_{\text{irr}}$, the subspace mask ratio $\rho$, the number of subspaces $k$, whether WISE-Retrieve.
**Output**: The final LLM model $f_{\Theta_T}$ after $T$ edits.
1: Generate $k$ random masks $\mathbf{M}_i, i \in [k]$ of ratio $\rho$; if WISE-Retrieve, copy the side memory several times;
2: **for** each edit $(\mathbf{x}_t, \mathbf{y}_t) \in \mathcal{D}_{\text{edit}}, t \in [T]$ **do**
3:      Edit $(\mathbf{x}_t, \mathbf{y}_t)$ in the corresponding memory subspace by $L_{\text{edit}} = -\log P_{W_{v'}}(\mathbf{y}_t|\mathbf{x}_t) + L_a$;
4:      Update the activation threshold: $\epsilon = \min(\epsilon, \Delta_{\text{act}}(\mathbf{x}_t))$;
5:      **if** All the $k$ subspaces of a side memory are full **then**
6:          Use Ties-Merge in Equation 8 to update the final side memory;
7:          **if** WISE-Retrieve **then**
8:              Move to another copy of side memory $\mathbf{W}_{v'}$;
9:          **end if**
10:     **else**
11:         **if** Current subspace $\mathbf{M}_i$ is full **then**
12:             Move to another subspace of side memory $\mathbf{M}_{i+1}$;
13:         **end if**
14:     **end if**
15: **end for**
16: **return** Obtain the final LLM model $f_{\Theta_T}$.

---

---

**Algorithm 2:** WISE Inference Stage

---

**Input**: The edited LLM model $f_{\Theta_T}$, the activation threshold $\epsilon$, the test dataset $\mathcal{D}_{\text{test}}$, whether WISE-Retrieve.
**Output**: The model's output.
1: **for** each query $\mathbf{x}_i \in \mathcal{D}_{\text{test}}$ **do**
2:     **if** WISE-Retrieve **then**
3:         Get the value of activation $\Delta_{\text{act}} = \|\mathcal{A}(\mathbf{x}_i) \cdot (\mathbf{W}_{v'} - \mathbf{W}_v)\|_2$ for each side memory and select the one with the maximal value of $\Delta_{\text{act}}$;
4:     **else**
5:         Get the value of activation $\Delta_{\text{act}} = \|\mathcal{A}(\mathbf{x}_i) \cdot (\mathbf{W}_{v'} - \mathbf{W}_v)\|_2$;
6:     **end if**
7:     **if** $\Delta_{\text{act}} > \epsilon$ **then**
8:         Use the side memory $\mathbf{W}_{v'}$ to generate the output as in Equation 6;
9:     **else**
10:        Use the main memory $\mathbf{W}_v$ to generate the output as in Equation 6.
11:    **end if**
12: **end for**

---

strategy that sets smaller Deferral radii during editing. Smaller Deferral radii will cause $\mathbf{x}_{e'}$ to fail to hit the codebook (the distance to the nearest *Key* is farther than its *Deferral radii*) but let $\mathbf{x}_{\text{loc}}$ successfully far away from the radii, resulting low generalization and high locality. Also, we observe that the Deferral radii method is not effective under any $\epsilon_{\text{init}}$; for all tested $\epsilon_{\text{init}}$ values of 1.0, 3.0, 10.0, and 500.0, they all have low generalization and high locality.

This suggests that in autoregressive LMs, the distribution of the last token cannot effectively represent semantics; whereas in encoder-only and encoder-decoder architectures, capturing semantic information through vector representation has been extensively studied [85–87]. This is consistent with the degree of generalization shown by GRACE when anchoring the T5 [88] Encoder layer. Some related works [89] also indicate that in autoregressive models, semantic similarity measures based on averages of output tokens underperform, recommending the use of score distributions over text continuations to represent semantic distances.

## B.2 Impact of Knowledge Merging Strategies for WISE

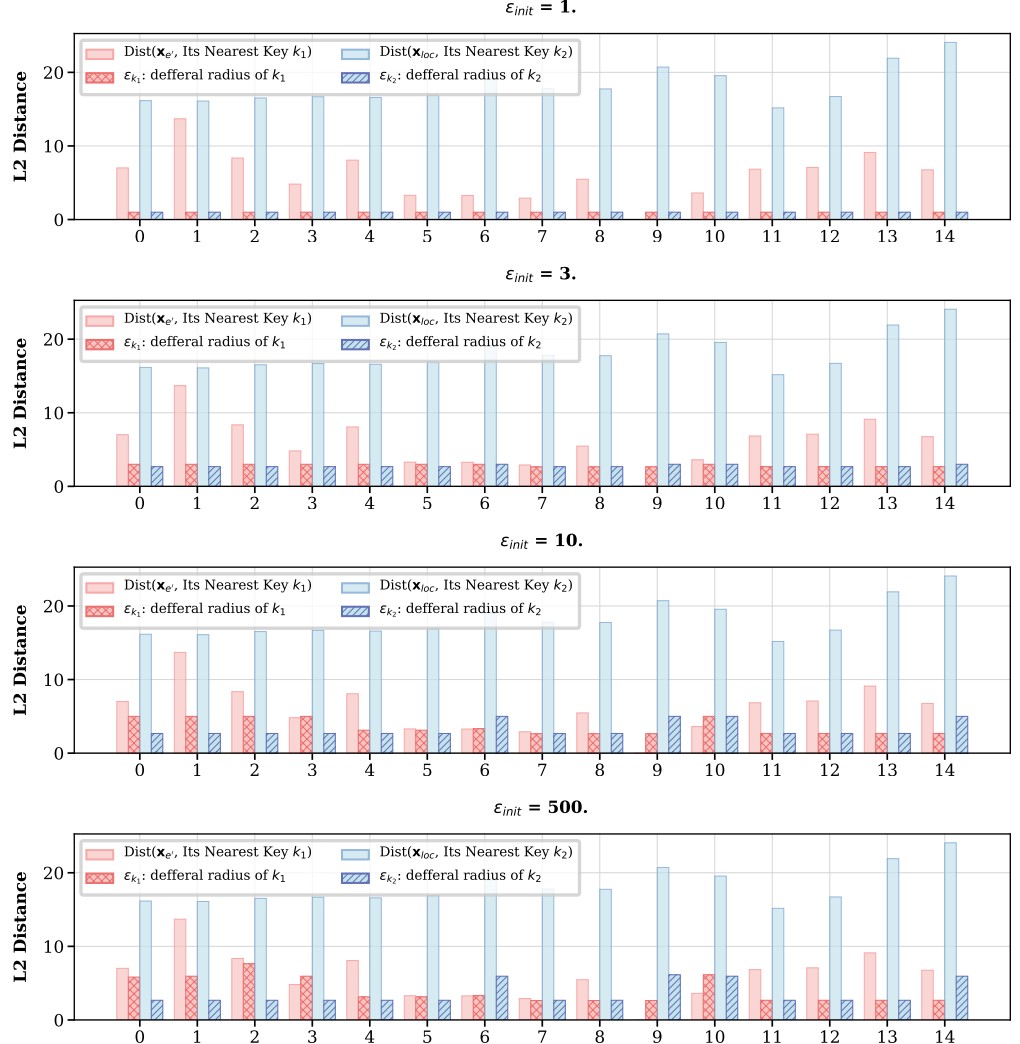

Figure 10: Investigation on the query $\mathbf{x}$ and its distance to the nearest Key $k$, as well as the *deferral radius* $\epsilon$ of that Key. Red and Blue respectively represent the paraphrase query $\mathbf{x}_{e'}$ and the unrelated query $\mathbf{x}_{\text{loc}}$, with the hatch representing the radius of the nearest Key. We observe that when conflicts occur (hit the codebook Key but with different Edit Target $\mathbf{y}_e$), the *deferral radius* $\epsilon$ decays exponentially. This results in GRACE being unable to encompass the paraphrase $\mathbf{x}_{e'}$ and maintain high locality, regardless of how $\epsilon_{\text{init}}$ is adjusted. ZsRE, `LLaMA-2-7B`.

Here, we conduct a more in-depth study of the knowledge merging strategies for WISE, exploring various merging approaches including (*i*) *Linear*, which uses a simple weighted average; (*ii*) *Slerp*, which spherically interpolates the parameters of two models; (*iii*) *Ties*, a component used in the main experiments of this paper that resolves merging disturbances through TRIM ELECT SIGN; (*iv*) *Dare*: which follows a Bernoulli distribution to delete redundant parameters and rescale the remaining ones; (*v*) *Dare_Ties*, which combines dare and the sign consensus algorithm of TIES; and (*vi*) *Sign*, an ablation component of Ties that addresses directional conflicts—all utilizing the official implementation from MergeKit [68] ‖. We randomly sample 100 edits from ZsRE, retaining a fine-tuned MLP every 50 edits (merging 2 MLPs). As shown in Table 11, we

Table 11: Varying Merging Strategy. ZsRE. `LLaMA-2-7B`.

| Methods | Rel. | Gen. | Loc. | Avg. |
|---|---|---|---|---|
| *Linear* | .63 | .61 | .93 | .72 |
| *Slerp* | .62 | .64 | .91 | .72 |
| *Dare* | .68 | .63 | .92 | .74 |
| *Dare_Ties* | .67 | .63 | .83 | .71 |
| *Ties* | **.85** | **.81** | .94 | **.87** |
| *Sign* | .80 | .76 | **.97** | .84 |

‖ https://github.com/arcee-ai/mergekit

observe that ignoring the direction of parameter updates (Linear, Slerp, Dare) leads to a significant decline in editing performance, underscoring the importance of addressing knowledge conflicts in overlapping parameters. The success of *Sign* also reaffirms this point. Meanwhile, the randomly masked knowledge shards exhibit a non-redundancy, indivisible nature. This is demonstrated by the significantly weaker performance of *Dare_Ties* compared to *Ties/Sign*, indicating that removing parameter updates can lead to the loss of edited knowledge or even potential "anchors".

## B.3 Analysis of Retrieving Top-1 Activation

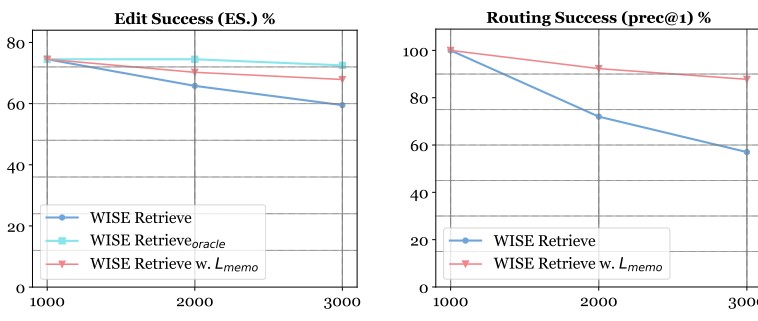

(a) Average of Rel. and Gen.  (b) Retrieval Acc. by Top-1 Activation

Figure 11: Comparing editing results of WISE-{Retrieve, Retrieve$_\text{oracle}$, Retrieve w. L$_\text{memo}$} when varying $T$. (a) shows the simple average of Rel. and Gen. (ES.), while (b) shows retrieval accuracy, i.e., whether the Top-1 Activation routes to the correct MLP (prec@1). X-axis: Num edits. ZsRE. LlaMA-2-7B.

WISE-Retrieve retains each knowledge-sharding memory and retrieves through Top-1 Activation. However, as shown in Table 6 and Figure 11b, the retrieval accuracy still has significant room for improvement; specifically, when $T$ reaches 3K, the accuracy of routing to the correct MLP drops to around 60%, indicating the specificity between side memories is insufficient. One possible reason is that when sampling the edits from a single dataset (ZsRE), the editing instances $(\mathbf{x}_e, \mathbf{y}_e)$ all belong to the same domain. This leads to some very similar instances being captured by multiple expert side memories (resulting in high activations for all side memories), introducing more retrieval failures.

Therefore, to improve the specificity of side memory and reduce the probability of routing errors, we attempt to add a new constraint $L_\text{memo}$ to Equation 5. For knowledge-sharding memory $\mathbf{W}_i$, we randomly replay instances $(\mathbf{x}_\text{m}, \mathbf{y}_\text{m})$ from the edit set $\mathcal{D}_{\mathbf{W}_j}$ of past shard $\mathbf{W}_{j, j \in [0, i-1]}$, ensuring that $\mathbf{W}_i$ remains inactive for $\mathbf{x}_\text{m}$:

$$L_a' = L_a + \underbrace{\max(0, \Delta_\text{act}(\mathbf{x}_\text{m}) - \alpha)}_{L_\text{memo}}, \quad \text{s.t. } \mathbf{x}_\text{m} \in \mathcal{D}_{\mathbf{W}_j}.$$

As shown in Figure 11b, this replay behavior increases the specificity between side memories, maintaining nearly **88%** retrieval accuracy at $T = 3K$. Figure 11a also shows that WISE-Retrieve w. $L_\text{memo}$ improves Edit Success (ES.) by **8.39%** compared to WISE-Retrieve, providing a promising direction for future work. With finer-grained activation management, we might be able to bridge the performance gap between Retrieve and Oracle.

## B.4 Case Study

In Table 12, we present bad cases of using WISE to edit the LLaMA-2-7B on the ZsRE dataset and mitigating these failures is critical for future work in model editing. We observe that in $i$) errors occur only in part of the tokens, and these errors constitute a large proportion of the bad cases, indicating that the edits have not been sufficiently fitted. $ii$) displays cases where the entire output is incorrect, and factual failures indicate difficulties in retaining memory of parameters for some rare entities (such as Persian $iia, iib$). $iv$) presents cases of generalization failure, for example in $ivd$), where the model answered "English" but did not fully follow the ground truth, indicating significant room for improvement in the accuracy of generalized edits. Meanwhile, in $iii$) we surprisingly find that even when WISE errs on the Edit Prompt, it can correctly answer its paraphrase $iiib$) *"The kind*

Table 12: **Failure cases of using WISE** to edit `LLaMA-2-7B`. ✘represents errors in part of the tokens, ✘represents complete output errors (i.e., factual failures), and ✓indicates the expected exact match.

| | Prompt | Edit Target | Post-Edit Output |
|---|---|---|---|
| *ia*) | By which person Lahti Town Hall has been designed? | Aki Kaurismäki | Wime Kaurismäki ✘ |
| *ib*) | *Which is the architect of Lahti Town Hall?* | - | Wime Kaurismäki ✘ |
| *ic*) | Which corporation was USS Leedstown (APA-56) created by? | Lockheed Shipbuilding | Leez Shipbuilding ✘ |
| *id*) | *Which company manufactures the USS Leedstown (APA-56)?* | - | Leez Shipbuilding ✘ |
| *iia*) | Which language is Garowe Principles written in? | Persian | Dutchian ✘ |
| *iib*) | *In what language does the monthly football magazine Garowe Principles report?* | - | Somian ✘ |
| *iic*) | What year was the service entry date for Panzer 58? | 1957 | 1953 ✘ |
| *iid*) | *What was the year Panzer 58 was commissioned?* | - | 1953 ✘ |
| *iiia*) | What was Gemma Bosini's range? | mezzo-srano | Wzo-srano ✘ |
| *iiib*) | *The kind of voice of Gemma Bosini is what?* | - | mezzo-srano ✓ |
| *iva*) | In which state is Qaleh Lan located? | Golestan Province | Golestan Province ✓ |
| *ivb*) | *What state is Qaleh Lan in?* | - | Lestan Province ✘ |
| *ivc*) | In which language Garowe Principles monthly football magazine reporting? | American English | American English ✓ |
| *ivd*) | *What language are Garowe Principles written in?* | - | English English ✘ |

*of voice of Gemma Bosini is what?"*. This indicates that WISE can handle contextual information correctly in some cases but falls short in specific editing instructions, suggesting that optimizing editing instructions (modifying the editing context) may be a direction for improvement.

## B.5 Importance of *Knowledge Anchor* When Merging Models

Table 13: Analysis of Merging *w.o.* and *w.* "knowledge anchor" (KA). $T = 1000$. ZsRE. `LLaMA-2-7B`.

| $\rho/k$ | *w.o.* KA | | | | *w.* KA | | | |
|---|---|---|---|---|---|---|---|---|
| | Rel. | Gen. | Loc. | Avg. | Rel. | Gen. | Loc. | Avg. |
| 2/0.30 | 0.76 | 0.72 | 1.00 | 0.83 | **0.79** | **0.73** | 1.00 | **0.84** |
| 2/0.50 | 0.74 | **0.73** | 1.00 | 0.82 | **0.77** | 0.72 | 1.00 | **0.83** |
| 3/0.33 | 0.72 | 0.68 | 1.00 | 0.80 | **0.75** | **0.71** | 1.00 | **0.82** |
| 5/0.20 | 0.64 | 0.61 | 1.00 | 0.75 | **0.73** | **0.68** | 1.00 | **0.80** |

Here, we discuss the effects of independent (ensured by non-overlapping masks) vs partially overlapping parameters within MLP subspaces on editing performance, as shown in Table 13. It is observable that, despite varying mask ratios $\rho$ and the number of subspaces $k$, partial overlap (w. KA) consistently outperforms independent configurations (w.o. KA) in terms of Reliability (Rel.) and Generalization (Gen.). For example, at $\rho/k$ of 5/0.20, there is a relative improvement of 9% and 7% respectively.

This demonstrates that the overlapping regions contribute as "anchors" for knowledge fusion, facilitating information transfer across different subspaces. Moreover, the shared parameters provide a natural regularization [90] mechanism, helping synchronize model behavior across different subspaces.

## B.6 Ablation Study of Random Prefix Token

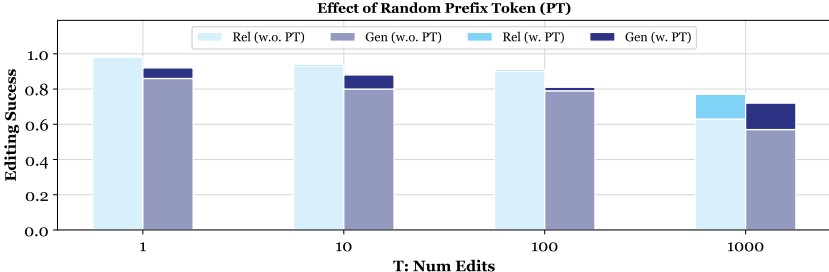

Figure 12: **Ablation studies on Random Prefix Token (PT) of WISE.** Light/Dark colors indicate the Editing Sucess w.o./w. PT addition. ZsRE. `LLaMA-2-7B`

As described in Section 3.1, we employ random prefix token augmentation to enable the editing knowledge to cope with various contexts. That is, for a single $\mathbf{x}_e$, it expands into $(\text{prefix}_i, \mathbf{x}_e)$. The prefix is derived from tokens that are randomly generated by the original LM $f_\Theta$, serving as an economical data augmentation method. We observe that the editing success rate is compromised (Figure 12). Specifically, for instance, at T=1000, Rel. and Gen. decreased by 0.15 and 0.17, respectively. By utilizing randomly generated prefix tokens, the model is able to learn a broader range of linguistic features, thereby exhibiting greater robustness in practical applications. We believe that access to the "data generator" can deepen the model's memory of editing samples.

### B.7 Parameter Efficiency

The key to lifelong model editing is maintaining constant or slowly increasing computational costs as the number of edits expands. Here, we provide a quantitative analysis using `LLaMA-2-7B` as an example. Suppose we select `model.layers[27].mlp.down_proj.weight` as side memory. In that case, the theoretically added parameters are $11008 \times 4096 \times 4 = 0.18$ GB, which accounts for

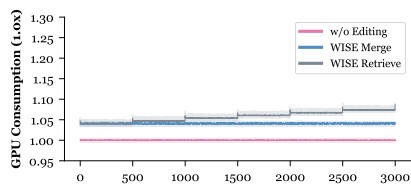

Figure 13: Computational costs.

0.64% of the original LLaMA's $7B \times 4 = 28$ GB (ignoring the VRAM required for input activations). As shown in Figure 13, in practice, WISE-Merge increases VRAM by 4% compared to the original `LLaMA` and remains constant over time. WISE-Retrieve, instead of merging, uses retrieval routing, meaning the computational cost increases over time, but this increase is gradual and can easily handle thousands or tens of thousands of inputs. Additionally, if we partially merge side MLPs (combining WISE-Retrieve and WISE-Merge), we can further reduce the computational demands of WISE-Retrieve.

## C  Proof of Theorem 2.1

**Theorem C.1 Subspace Overlap.** *Generate $k$ memory subspaces $\mathbf{W}_{v'}^i, i \in [k]$ by random mask with 1's ratio $\rho$, so each memory has $\rho \cdot |\mathbf{W}_{v'}|$ active trained parameters. For any two subspaces $\mathbf{W}_{v'}^i$ and $\mathbf{W}_{v'}^j$ $i \neq j; i, j \in [k]$, there are $\rho^2 \cdot |\mathbf{W}_{v'}|$ active parameters that are overlapped. For all $k$ subspaces, there are $\rho^k \cdot |\mathbf{W}_{v'}|$ overlapped active parameters.*

*Proof:* We aim to prove the Subspace Overlap theorem by induction.

Let $\mathbf{W}_{v'}^i$ represent the $i$-th memory subspace generated by a random mask with a sparsity ratio of $\rho$, where $i \in [k]$. Each memory subspace $\mathbf{W}_{v'}^i$ contains $\rho \cdot |\mathbf{W}_{v'}|$ active trained parameters.

We start by considering the case of two memory subspaces, $\mathbf{W}_{v'}^i$ and $\mathbf{W}_{v'}^j$, where $i \neq j$ and $i, j \in [k]$. Let $P(\text{parameter sampled}) = \rho$ be the probability that a parameter is sampled in one mask generation event.

1. For a single mask generation, the probability that a specific parameter is sampled is $\rho$. We denote this probability as $P(\text{sampled}) = \rho$.

2. Considering two independent mask generation events, the probability that the same parameter is sampled in both masks is the product of their individual probabilities, i.e., $\rho^2$. This is derived from the independence of the events. Mathematically:
$$P(\text{sampled in both masks}) = P(\text{sampled}) \times P(\text{sampled}) = \rho \times \rho = \rho^2.$$

3. Extending this logic, for $k$ independent mask generation events, the probability that a specific parameter is sampled in all $k$ masks is $\rho^k$. Mathematically:
$$P(\text{sampled in all } k \text{ masks}) = \underbrace{P(\text{sampled}) \times P(\text{sampled}) \times \cdots \times P(\text{sampled})}_{k \text{ times}} = \rho^k.$$

Now, let's calculate the number of parameters overlapped in two random masks:

The total number of parameters in $\mathbf{W}_{v'}$ is $|\mathbf{W}_{v'}|$.

Thus, the number of parameters overlapped in two random masks, $\mathbf{W}_{v'}^i$ and $\mathbf{W}_{v'}^j$, is $\rho^2 \cdot |\mathbf{W}_{v'}|$.

Extending this to $k$ random masks, the number of parameters overlapped in all $k$ masks is $\rho^k \cdot |\mathbf{W}_{v'}|$.

This concludes the proof.

$\square$

# D  Detailed Related Works

**Memory and Knowledge Injection of LLMs**  The memories of LLMs can be divided into long-term (episodic) memory and working memory (short-term) [24, 25, 27]. Long-term memory refers to the knowledge stored in the model's parameters, which can be updated by (re)pretraining [53], finetuning [59], and model editing [14]. Working memory is stored in sustained activations/representations of neurons, which will be awakened during inference time [24]. In-context learning (ICL) is a kind of working memory [60], also along with retrieval-based editing methods like GRACE [10]. How to reinforce memory and inject/update knowledge for LLMs is a fundamental question [28, 61, 62]. ICL or finetuning? Different works show different conclusions. In [62], the authors find that few-shot finetuning is more generalizable than ICL, especially for out-of-distribution data. In [61], the authors contrast finetuning with retrieval-augmented generation (RAG) in terms of knowledge injection and find that RAG is better in most cases, and combining both will produce the best results. However, finetuning and pretraining are computation-expensive [13, 10] and usually suffer from catastrophic forgetting [63] and overfitting [64]. For ICL and RAG, the working memory is sometimes not controllable, the model may not follow the information of the contexts [24], and the context window is limited [91, 92], and there are works addressing these issues by training controllable ICL [24], long-context [91, 92], and recurrent memory architecture design [28]. SPALM is proposed to add language models with storage modules that resemble both working and long-term memories [27].

**Model Editing of LLMs**  Model editing can be summarized as the following lines of research. *Constrained finetuning:* Preliminary model editing uses constrained finetuning to update parameters based on new examples [93, 94]. *Locate-and-edit:* ROME [18] locates the factual associations in autoregressive LLMs and conducts accurate and efficient edits by taking MLPs as key-value memories. Then, MEMIT [19] extends ROME from single-editing to mass-editing. COMEBA-HK [95] identifies the Local Editing Scope and extends MEMIT for sequential editing. In addition, T-Patcher [11] targets the last feed-forward layer of LLMs, adding an additional neuron for each edit. *Meta learning:* Recent meta-learning methods use hypernetworks for aiding editing. MEND [31] learns a hypernetwork that can decouple the finetuning gradients into the gradient updates that generalize the edits and won't damage the performances on unrelated inputs. To remedy the cancellation effect of MEND, MALMEN [15] uses hypernetwork to produce the weight shifts of editing and formulates the weight shift aggregation as the least square problem. *Retrieval-based methods:* Instead of directly editing the model parameters, retrieval-based methods aim to improve the working memory of LLMs to enable model editing. IKE [96] uses context-edit facts to guide the model when generating edited facts. DeCK [97] employs contrasting knowledge decoding, which enhances the confidence of in-context-based editors in the edited facts. SERAC [32] (a modified version dubbed as DEFER [10]) records edit items in a file and trains additional scope classifier and counterfactual model to detect, retrieve, and generate the edit-related results. Though the editing retriever and generator are neural networks, they are too small to have the power of LLMs. GRACE [10] adopts a discrete codebook of edits for retrieving and replacing the edits' layer representations during inference. From single editing [18] to mass editing [15, 19], and from static editing to sequential [11] (continual) or lifelong editing [10], model editing is developing to meet more realistic demands.

**Continual Learning**  Continual learning [98, 99] tackles the catastrophic forgetting problem in deep learning models with new knowledge [100], and recent research has focused on various methods in this area. One such method is continual finetuning, where LLMs are refined over time with the arrival of new instances. For instance, a comprehensive study by [101] explores continual finetuning extensively. However, it has been observed that regularizing finetuning with continual learning techniques such as Elastic Weight Consolidation [20], Experience Replay [102], and Maximally Interfered Replay [103] can lead to a rapid decay in performance on previous tasks, although it aids in retaining some memory of past inputs. This suggests that editing, as opposed to vanilla continual finetuning, presents unique challenges, especially considering that edits are unlikely to be evenly distributed [104]. One promising direction within the realm of continual learning is the adoption of key-value methods, inspired by advancements in computer vision [105, 106]. Recent studies have showcased the effectiveness of continual prompt-learning for NLP [107, 108], particularly in applications like text retrieval [109]. Notably, discrete key-value methods have been shown to excel in handling shifting distributions [110], with some recent efforts extending their application to question answering [111]. These methods cache values to ensure that inputs remain within the distribution for downstream encoders, thus facilitating the incorporation of longer-term memory, provided there are adequate computational resources.

