# OpenReview forum: "WISE: Rethinking the Knowledge Memory for Lifelong Model Editing of Large Language Models"
_NeurIPS.cc/2024/Conference — NeurIPS 2024 poster_

### Official Review · Reviewer_fviW · 2024-07-11

**Soundness:** 3
**Presentation:** 3
**Contribution:** 3
**Rating:** 7
**Confidence:** 4

**Summary:**

Introduces a new technique, WISE, to edit/insert facts in a autoregressive language model. The proposed approach gives a balanced performance in terms of reliability, generalization, and locality; improving upon the existing methods. WISE edits facts in a side-memory that is equivalent to one of the MLP down projection layers. A router is trained to decide if a specific query should be routed to the side-memory or the corresponding layer in the main LM.

**Strengths:**

* WISE is a novel approach that balances reliability, generalization, and locality while editing facts in a LM.
* Clear motivation and solid technical contributions.

**Weaknesses:**

Needs more clarity on some of the technical details. See the questions below.

**Questions:**

1. The activation $\textbf{a}$ used in routing correspond to which token in the prompt? Is it the subject last token like ROME/MEMIT or the last token of the prompt like GRACE?
    * One of the reason GRACE achieves poor generalization is because it edits the last token of the prompt and thus very sensitive to the prompt paraphrases. The notation used in the paper suggests that WISE also uses the last token of the prompt (clarify this if I am wrong). However WISE seems to achieve an impressive generalization. Is that because the parameters of $\mathbf{W}_{v'}$ are directly edited for better routing? Also, does WISE use only one prompt $x_e$ to edit a specific fact, or does it use multiple paraphrases of $x_e$ during the training phase?

2. Parameter Overlap: Theoren 2.1 seems to counts parameter overlap as the number of parameters that are common between *all* the $k$ side-memories as $\rho^k|\mathbf{W}|$. But if a parameter is common between only two side-memories, it is also a overlap, right? And, in that case, the number overlaps would be $\Big(\binom{k}{2}\rho^2 - \binom{k}{3}\rho^3 + \binom{k}{4}\rho^4 \cdots \Big) \cdot |\mathbf{W}|= \sum_{i=2}^k (-1)^i \binom{k}{i}\rho^i \cdot |\mathbf{W}_{v'}|$.
    * I also didn't follow the point on the overlapping playing the role of "*anchors*" for knowledge merging, even after refering to Appendix B.5, which just restates the same point. Can you please clarify this?
    * I also didn't follow the claim made in line 191-192 => *"different shards of edits have different random masks, and due to the (sub)orthogonality of random masks, different shards will not conflict with each other"*.

3. Although WISE claims to be sequential (line 222), it doesn't seem to be the case, as you expect access to $n$ edits to distribute among $k$ shards (line 179-180). Or, is the loss $L_{edit}$ optimized for each of the edits sequentially? Also, how do you perform the merging if you don't have access to all the edits beforehand? If WISE was truly not done sequentially, then it should only be compared with methods that permit parallel/batch edits.

4. As you increase the number of edits $T$, do you also increase the number of sharts $k$? Or, do you always keep $k = 2$ reported in Table 9? Also since the title of the paper focuses on *Lifelong* model editing, I wonder if it is reasonable to increase $k$ rather than choosing a fixed $k$.

**Limitations:**

Limitations have been adequately addressed in the paper.

---

> ### Author Rebuttal · Authors · 2024-08-05
>
> # Response to Reviewer fviW
>
> Thanks for your valuable feedback. We appreciate the opportunity to address your concerns.
>
> > 1. Response to “The activation $\mathbf{a}$ used in routing corresponds to which token in the prompt?”
> >
>
> As shown in Equation 3, the activation used for classification is derived from the entire prompt, meaning $\Delta_{\text{act}}$ represents the **average activation across all prompt tokens**. This increases the probability of correctly classifying the paraphrase $x_e'$.
>
> > 2. Response to “Impressive generalization of WISE.”
> >
>
> Thanks for the comment, we suppose there are three key reasons behind WISE's impressive generalization:
>
> - By averaging activations across all prompt tokens, we enhance the model's ability to identify the scope of edits, allowing paraphrase $x_e'$ to pass through side memory while unrelated inputs $x_i$ pass through main memory.
> - $\mathbf{W}_{v'}$ is directly modified (Long-term Memory), copying FFN parameters from the original LLM and memorizing new editing samples with a small portion of parameters. During inference, side memory and earlier layers are packaged together, enhancing overall representation capabilities beyond activation-based methods.
> - We employ random text augmentation (avoiding leakage of paraphrase inputs). Initially utilized in ROME: For a single $x_e$, it expands into (prefix_i, $x_e$). The prefix is derived from tokens randomly generated by the original LM, serving as an economical data augmentation method.
>
> > 3. Response to the issue of “parameter overlap”.
> >
> - **Why $k$-memories’ overlap:** Because during model merging, we merge several parameter subspaces into one, not pair-wise merging, we consider the $k$-memories’ overlap.
> - **Explanation of the “anchors”:** For vanilla mode merging, full models are fine-tuned to merge, so the overlap is 100% in the original Ties-Merge. While in our random subspace, only a small proportion ($\rho$) of parameters are updated in a subspace. Therefore, when conducting model merging, it is necessary to have overlapped parameters to rescale and rectify (e.g., trim/select sign in Ties-Merge) different model pieces.
>     - We also made an ablation study. In Table 12, "w. KA" (generate random subspaces with replacement) consistently outperforms "w.o. KA" (generate disjoint subspaces by random sampling without replacement), demonstrating that anchors improve performances.
> - **(Sub)orthogonality of random masks:** Since the proportion of subspace overlap is small (e.g., 0.03), most of the parameters are disjoint, which means that the knowledge in different subspaces will cause little conflicts; here, we note this as sub-orthogonality. There is a tradeoff: avoiding knowledge conflicts requires subspaces to have most parameters disjoint, but the parameters cannot be totally disjoint, otherwise the model merging will be poor, so we intuitively observe nuances in the tradeoff where 0.03 of overlap can realize the best result across different $\rho$ and $k$.
>
> > 4. Response to “Does it use multiple paraphrases of $x_e$ during the training phase?”
> >
>
> Sorry for the confusion caused, we will elaborate on this point as follows.
>
> - As mentioned above, we employ LM-generated token augmentation during the training phase to enable the LM to remember knowledge updates across different contexts. This approach is also employed in ROME, MEMIT, FT, etc., as documented in the supplementary material.
> - However, we acknowledge the oversight regarding GRACE, which accepts only bs=1 inputs. In the rebuttal phase, we've completed these experiments, applying the 11 "augmented data" instances generated by LMs to GRACE. When epsilon_init = 500.0, the results of GRACE are shown in **Table C** of the ***Rebuttal PDF*** **in the general response.**
>     - Overall, GRACE demonstrates generalization (e.g., Gen. is 0.28 higher at T = 1 compared to Table 1). However, at T = 1000, due to the inability of the last token (as mentioned in Appendix B.1) to represent semantics, its generalization performance remains inadequate. We will update the main table with GRACE's experimental results.
> - We conducted an ablation study on random text augmentation, the final performance is in **Figure B** of the ***Rebuttal PDF.***
>     - We observe that **the editing success rate is compromised**. We believe that access to the "data generator" can deepen the model's memory of editing samples, and this constant overhead is worthwhile. This finding will also be added in the Appendix.
>
> > 5. Response to “Perform merging though we don't have access to all the edits beforehand.”
> >
>
> In Figure 5, we discuss knowledge density, finding optimal $\rho^k$ consistently near 0.03, with good performance within the interval $\rho=[0.3, 0.5]$ (exhibiting robustness). This suggests that even without prior knowledge of incoming knowledge count (N), current experiments indicate that 0.2 FFN parameters can accommodate at least 500 edited samples. When "mask memory exhaustion" occurs, we can allocate new mask parameters to store new knowledge. Using retrieve when knowledge isn't full and merging as needed to save memory, achieves true lifelong model editing.
>
> > 6. Response to “The number of shards k when T increases.”
> >
>
> We do not only test the editing performance for k=2. As shown in Figure 5, k=2 and $\rho$=0.2 exhibit optimal performance at the scale of T=1000, which is why it was reported in the main table. Figure 5 also presents results for k=[2,3,4,5,6], revealing interesting conclusions about knowledge density. Furthermore, in Table 6, we extend the number of editing instances to 3K, resulting in 6 merges (3000 / 500). Please refer to General Response 1, where we discuss that as k increases, single side memory has its limited knowledge capacity. Therefore, combining WISE-Retrieve with timely merges and fine-tuning after accumulating sufficient mistakes can meet the requirements of real applications for deployed online models.

---

> > ### Comment · Reviewer_fviW · 2024-08-12
> >
> > I appreciate the authors for their detailed response. I would like to bump my rating to 7. Congratulations on your fine work!

---

> > > ### Author Response · Authors · 2024-08-12
> > > **Thanks for the post-rebuttal response**
> > >
> > > We are grateful for your efforts in engaging with our work and your support in raising your score. Your assessment has been invaluable in refining our work and clarifying the key aspects of our research. We will further improve the experiments related to random prefix tokens and clarify that WISE can perform edits sequentially in the updated version of the manuscript.
> > >
> > > Thank you once again for your time and consideration.

---

### Official Review · Reviewer_pT3T · 2024-07-12

**Soundness:** 3
**Presentation:** 2
**Contribution:** 3
**Rating:** 7
**Confidence:** 3

**Summary:**

The paper studies the problem of editing (updating) knowledge of LLMs in a lifelong learning scenario. Authors design WISE, a multi-level memory system designed to store updates to the model. The proposed design contains of the main memory and a number of side-memories, with a mixture-of-expert-like router to choose between them. Authors also propose a procedure for merging the main (mid-term) memory into side-memories effectively committing it to the long-term memory. Authors evaluate the proposed technique on small variants of popular LLMs, achieving good average performance on ZsRE and a number of other datasets (SelfCheckGPT, Temporal). The paper also contains several side-analyses such as visualizing the router behavior, scaling to 3K edits, and speed benchmarks.

**Strengths:**

- The problem of updating LLM parameters is important and a practical improvement in this direction can reduce the total cost of using LLMs (both economic and environmental)

- The paper contains a variety of experimental configurations and baselines, including several models of different capability (and creation date), different datasets, and numerous baseline algorithms. This would normally be expected of a NeurIPS submission, but sadly, often not the case. The only possibly unexplored dimension is model size: testing with 6-7B models may not reveal some caveats that arise only in larger ones.

- As a minor but helpful touch, the provided code contains clear instructions on running the code and well-defined dependency versions. This helps future researchers reproduce and build on top of this work.


The paper is also generally well written and reasonably well structured, though I got the impression that authors tried to squeeze a lot of information into few pages. If this paper ends up accepted, I respectfully ask that authors reduce the total negative (vertical) space used in formatting by exiling some of the less important analyses to appendix or, if all else fails, by using the extra content page granted for the final version.

**Weaknesses:**

I have two main concerns, though they are not significant ones.

First WISE is a rather complicated system with a lot of moving parts: router type, merging strategy, main/side memory sizes, where to introduce this module into an LLM, and how many times, how best to allocate the memory size between components. Authors provide some ablation analysis (e.g. Appendix B.2 and below), but many of these ablations are missing.

My second (very minor) concern is that authors experiment only with small LLMs (sic), which leaves out the possibility that WISE behaves unexpectedly with larger and higher-capability ones (at the minute of writing this, Llama-3-70B, Qwen-2-72B, Nemotron et al.).

**Questions:**

I do not have any insightful questions, so I will instead use this section for minor comments.

### Typos, missing citations, minor concerns

> L389 Malicious users may attempt to edit LLMs to propagate hate, highlighting the need …

There are numerous other scenarios for a potential misuse of this technology. To name a few:
censorship, particularly for non-democratic state actors
misinformation, of a non-hate-inducing kind

For the record: i do NOT mean that the paper requires an ethical review. Most of these attack vectors are already possible with prior work. It would be unrealistic to expect a full sister study on ethics. Fortunately, we have AI ethics researchers.


> L34 should satisfy the following properties [ 14 , 15, 11 ]

I believe this terminology were originally introduced earlier in Sinitsyn et al (2020) [ https://openreview.net/forum?id=HJedXaEtvS ] and De Cao et al (2021) [ https://aclanthology.org/2021.emnlp-main.522/ ], including for language models. Though the term “LLM” specifically did not exist back then.

> L25 parameters, computes, and data

While using “compute” as a noun is not yet well studied, I have seen it mostly used as an uncountable noun (like “data” instead of “datas”). Please use your own judgment though.

> L98 (definition formula for D_edit)

To the best of my knowledge, using | for nested definition of Xe,Ye is rarely used (or understood) by ML practitioners. Consider defining them separately




Overall, if this paper ends up accepted, I respectfully ask that authors reduce the total negative (vertical) space used in formatting by exiling some of the less important analyses to appendix or, if all else fails, by using the extra content page granted for the final version.

**Limitations:**

To the best of my knowledge, authors have sufficiently addressed the limitations of their work.

---

> ### Author Rebuttal · Authors · 2024-08-05
>
> # Response to Reviewer pT3T
>
> Thanks for your valuable feedback. We appreciate the opportunity to address your concerns.
>
> > 1. Response to “many of ablations are missing.”
> >
>
> Thanks for the comment. All components of WISE can be summarized as router, merging strategy, locating side memory, and side memory sizes. Indeed, except for the merging strategy detailed in Appendix B.2, these components are either ablated (if applicable) or analyzed in the main body of the paper. We discuss the independent contributions of these components as follows:
>
> - **Router**
>     - As shown in Figure 3, we present visualizations of WISE router activations. The purple horizontal line in the figure represents the minimum activation threshold $\epsilon$ recorded during the editing phase. Nearly all irrelevant queries (blue) show lower activations, while inputs within the editing domain ($x_e$ or its paraphrased form $x_e'$) exhibit higher activations (red), underscoring the efficacy of the router. Without the router strategy, all inputs either pass solely through the main or side memory. To further validate its effectiveness, we conduct additional ablations with $L_a$. WISE's performance on ZsRE is shown in **Table A** of the ***Rebuttal PDF*** **in the general response**.
>     - Observing the expected **decrease in Loc.** w.o. $L_a$ , such as dropping from 1.00 to 0.72 at T=1000, reveals the router's effectiveness in identifying editing scopes, minimizing side effects, and retaining a substantial amount of pre-training knowledge.
> - **Merging strategy**
>     - We provide detailed ablations in Appendix B.2. Table 10 shows that simple linear/slerp interpolation performs poorly, whereas Ties and Sign excel. It also effectively demonstrates a) the necessity of non-redundancy of masked knowledge shards; b) we must resolve conflicts during merging to ensure maximal retention of all past edited samples.
> - **Main/side memory sizes**
>     - The main memory is part of the original LLM, hence its size is not under our control. Side memory replicates a layer of FFN from the LLM, initialized with its weights. Theoretically, it requires 0.64% of the original LLM's parameters (using LLaMA-2-7B as an example).
> - **Where to introduce this module into an LLM, and how many times**
>     - *Where to introduce*: Extensive literature [1,2] suggests LLMs encode high-level semantics in mid-to-late layers and handle more advanced linguistic phenomena. As shown in Figure 4, we select early, intermediate, mid-to-late, and late stages, finding superior editing performance in mid-to-late layers, possibly due to the formation of more complex semantics/knowledge. Overall, we provide guiding recommendations for "Where to introduce." Additionally, we attempt to generalize to the 13B model, as shown in **Table B** of the ***Rebuttal PDF***, confirming these findings.
>     - *How many times*: In Figure 5, we discuss knowledge density (closely related to "how many times"), finding optimal $\rho^k$ consistently near 0.03, with good performance within the interval $\rho$ =[0.3, 0.5] (exhibiting robustness). This suggests that even without prior knowledge of incoming knowledge count (T), current experiments indicate that 0.2 FFN parameters can accommodate at least 500 edited samples. When "mask memory exhaustion" occurs, we can allocate new mask parameters to store new knowledge. Using retrieve when knowledge isn't full and merging as needed to save memory, achieves true lifelong model editing.
>
> > 2. Response to “WISE's behavior with larger and higher-capability LMs (e.g. LLaMA-3-70B, Qwen-2-72B)”.
> >
>
> Thanks for your suggestions. In this paper, we explore the latest (at least at the submission stage) and popular LLMs (LLaMA-2-7B, Mistral-7B), observing significant and consistent improvements with WISE across multiple experimental settings. Regarding LLMs of the 70B scale, we lack sufficient resources to conduct such experiments. However, to the best of our abilities, we attempt to observe WISE's performance on **LlaMA-2-13B-chat**, as shown in **Table B** of the ***Rebuttal PDF*** (choosing the mid-to-late layer: `model.layers[34].mlp.down_proj`):
>
> Experimental results validate WISE's efficacy on the 13B-chat model, even surpassing editing performance on the 7B model at T=1000. This enhances WISE's scalability across model sizes, and we plan to further refine experiments and attempt editing larger 70B models. Once again, thank you for your suggestions.
>
> > 3. Response to “Typos, missing citations, minor concerns”
> >
>
> Many thanks for your kind attention and carefulness. We make the following responses.
>
> - Ensuring safe and responsible practices in WISE is of paramount importance. Model editing, which can suppress harmful language generation [3-5], could help address these concerns.
> - We have added citations for Sinitsyn et al. (2020) and De Cao et al. (2021) in the updated version of the paper and removed the incorrect term "compute" in L25.
> - We have also added a reference to Table 7 when introducing D_edit, which provides specific cases of x_e and y_e. Thank you for pointing out these issues.
> - We will remove all \vspace in the CR phase to ensure the paper's formatting is more consistent.
>
> ---
>
> [1] Tianjie Ju, et al. ”How Large Language Models Encode Context Knowledge? A Layer-Wise Probing Study.” *LREC-COLING 2024*
>
> [2] Yung-Sung Chuang, et al. ”DoLa: Decoding by Contrasting Layers Improves Factuality in Large Language Models.” ICLR 2024
>
> [3] Xinwei Wu, et al. “DEPN: Detecting and Editing Privacy Neurons in Pretrained Language Models.” EMNLP 2023
>
> [4] Mor Geva, et al. “Transformer Feed-Forward Layers Build Predictions by Promoting Concepts in the Vocabulary Space.” ACL 2022
> [5] Mengru Wang, et al. “Detoxifying Large Language Models via Knowledge Editing." ACL 2024

---

> ### Comment · Reviewer_pT3T · 2024-08-14
> **Response**
>
> I apologize for responding late and thank authors for a detailed response. Authors have answered my questions in full and suggested reasonable updates to the final version of the paper. I am keeping my score as is (7), which is to say, I recommend that the paper gets accepted.

---

### Official Review · Reviewer_5h8i · 2024-07-14

**Soundness:** 3
**Presentation:** 3
**Contribution:** 3
**Rating:** 6
**Confidence:** 4

**Summary:**

This paper proposes WISE which uses a side memory and some model merging techniques to perform lifelong knowledge editing.

**Strengths:**

1. The paper is well-written and easy to follow.
2. The experiments are somewhat comprehensive.
3. The paper is working on the continual editing problem, which is important.

**Weaknesses:**

1. **Ever-expanding memory**: As mentioned in line 212-213, "One single side memory has its limited knowledge capacity". Thus, to perform lifelong knowledge editing, it seems that infinite side memories would be needed. In this sense, lifelong knowledge editing is still not solved as with limited side memories, only a limited number of edits can be performed. As the ever-expanding external memory is needed, why do we need to store all this knowledge in the memory instead of just storing the knowledge in the form of raw text and performing knowledge retrieval from this knowledge base?
2. **Experimental Results**: In the paper MEMIT [1], they scale the batch editing to 10000 examples. There are around 20000 examples in zsRE dataset. However, in the main table (Table 2) only up to 1000 edits are studied.
3. **Sequential Editing Cases**: Two possible scenarios in sequential editing are: (1) There would be conflicting knowledge overtime. For example, The president of some country may continue changing every a few years. How does WISE perform in this case compared to others? (2) Multi-hop edits as discussed in [2]. However, it seems that both of these situations are not discussed in the paper.

[1] MASS-EDITING MEMORY IN A TRANSFORMER.
[2] MQuAKE: Assessing Knowledge Editing in Language Models via Multi-Hop Questions.

**Questions:**

1. **Experimental Results**: In MEMIT[1], they achieve 96.7 on Efficacy and 89.7 on Paraphrase (see Table 1) with 10,000 edits, but in Table 6, MEMIT-MASS has 0.64 in Rel. and 0.58 in Gen. under 2000 edits and 0.58 in Rel., 0.53 in Gen. under 3000 edits. What happened to the model to contribute to the discrepancy between the current results and the results reported in the paper?

**Limitations:**

The limitation of ever-expanding memories may need to be mentioned in the limitation section.

---

> ### Author Rebuttal · Authors · 2024-08-05
>
> ## **Response to Reviewer 5h8i**
>
> We thank the reviewer for acknowledging that our paper is well-motivated and the experiments are comprehensive. We kindly address your questions as follows.
>
> > 1. Response to “Why not store the knowledge in the form of raw text and perform knowledge retrieval (RAG)?”
> >
>
> Thanks for your valuable comments. **We kindly refer to the general response for the concern of “ever-expanding memory”.** Then, we will elaborate on the following aspects and explain why WISE is irreplaceable compared to raw text retrieval:
>
> 1. **Lifelong Perspective on Knowledge Update**
>     - **Fact Reasoning/Utilization**: Let us consider a simple example: Suppose we aim to edit A: "change *Oliver Twist*'s author to *Shakespeare*" and then to edit B: "change *Shakespeare*'s nationality to *French*". If we query "In which country was *Oliver Twist* written?", a similarity-based retrieval would likely only retrieve A. Using raw text cannot establish generalized knowledge associations both externally and within the model, resulting in weak generalization and poor reasoning abilities. Even recent literature has begun using knowledge graph construction for GraphRAG to mitigate this dilemma, but it is inevitably limited by information extraction technology's inability to establish precise associations.
>     - **Adaptive Chameleon or Stubborn Sloth [1]**: LLMs demonstrate a strong confirmation bias toward parametric memory when faced with both supportive and contradictory evidence to their parametric memory [1]. This means that RAG faces additional challenges in optimizing raw text to be coherent and convincing.
>     - **Efficiency of Inference**: Retrieval itself is a massive engineering system; common embeddings, retrieval, rankers, and synthesis all cause significant delays in online systems.
> 2. **Synergy Between WISE and Raw Text Retrieval**
>     - WISE and RAG are independent; WISE updates internal model knowledge as parametric knowledge, which can be combined with RAG (non-parametric knowledge) to achieve more robust knowledge updates.
>
> To validate these points, we conduct comparative experiments on the MQuAKE dataset (evaluation on $x_e$'s multi-hop query). For **ICE** (In-Context Editing, i.e., RAG), we choose `all-mpnet-base-v2` and retrieve the Top-1 cosine similarity editing fact to construct the prompt: `[Updated Information]: \n {retrieved-editing-fact} \n [Query]: {query}`. Results are shown in **Figure A** of the ***Rebuttal PDF*** **in general response**.
>
> We first observe that **WISE + ICE consistently outperforms** WISE or ICE alone, proving our point that combining the two achieves better multi-hop reasoning performance. Secondly, as T increases to 1000 and 2000, **WISE demonstrates the advantage of utilizing editing facts x**, combining and associating knowledge to enhance reasoning.
>
> > 2. Response to “MEMIT scales the batch editing to 10000 examples.”
> >
> - As shown in Appendix A.2, MEMIT-MASS does not belong to sequential edits but rather to parallel/batch edits (reviewer fviW also points out this), **losing the capability for on-the-fly repairs.** The main table also demonstrates MEMIT's sequential editing performance, akin to ROME, where the model is fully compromised by T=100.
> - Lifelong editing was first proposed in GRACE. We follow GRACE's settings and dataset, reporting editing performance at T=1000 in the main table, with analysis extending to T=3000 in Section 3.3. Comprehensive analysis from Table 6 and Figure 11 shows that WISE is capable of scaling up to more edited samples. We recommend using WISE to respond to each mistake online promptly. After accumulating a sufficient number of mistakes, fine-tuning the original model on all accumulated mistakes allows for the removal of side memories. Then, WISE can be reused for on-the-spot repairs. The ability to scale to thousands/ten thousands of edits can meet the requirements of real applications for deployed online models.
>
> > 3. Response to the issue about “the discrepancy between the current results and the results reported in the MEMIT paper”.
> >
>
> This discrepancy arises from **different metrics yielding different results**. We follow GRACE's metrics. As shown in Equation 9 of our paper: $\text{Rel.} = \frac{1}{T}\sum\limits_{t=1}^{T} \mathbb{1}(f_{\Theta_{T}}(x_e^t) = y_e^t)$. In MEMIT paper 5.2.2, Efficacy Success (ES) is defined as the probability of the new object $o'$ being greater than the original object $o$: $E[P_{G} [o_i' | p(s_i, r_i)] > P_{G}[o_i | p(s_i, r_i)]]$. We argue that determining editing success based on **real output alignment is stricter (thus, smaller value)** and more aligned with real-world scenarios.
>
> > 4. Response to the issue about “Knowledge Conflict in sequential editing”.
> >
>
> In Appendix B.3, we discuss the accuracy of Retrieving Top-1 Activation, demonstrating the efficacy of WISE in identifying the correct side memory. We propose an inspection module as a remedy for WISE:
>
> - First, for a given editing knowledge, use routing activation to *check whether it matches past editing sequences and identify the corresponding side memory*.
>     - If so, cover previous knowledge through re-editing this side memory using current new knowledge in a tiny subspace (e.g., selecting a 0.01 mask parameter), which will rewrite the previous conflicting knowledge while minimally affecting other editing knowledge.
>     - Otherwise, conduct vanilla WISE steps, i.e., introduce new parameter subspaces for inserting new knowledge.
>
> > 5. Reponse to “Editing results of Multi-hop query (MQuAKE dataset).”
> >
>
> We have supplemented these results in **Figure A** of the ***Rebuttal PDF*** **in general response**. The discussion regarding these results is shown in the second response in your thread. We will further refine this experiment and update the manuscript.
>
> ---
>
> [1] Jian Xie, et al. “Adaptive Chameleon or Stubborn Sloth: Revealing the Behavior of Large Language Models in Knowledge Conflicts.” ICLR 2024

---

> > ### Comment · Reviewer_5h8i · 2024-08-13
> > **Response to the rebuttal**
> >
> > Thank the authors for the detailed responses!
> >
> > I think my concerns are addressed, I have raised my score accordingly.
> >
> > Thanks!

---

> > > ### Author Response · Authors · 2024-08-14
> > > **Thank you note to Reviewer 5h8i**
> > >
> > > We sincerely thank you for your engagement and kind acknowledgment of our efforts to address your insightful comments. We are thrilled that your questions have been answered, and we deeply appreciate the increased score.

---

### Official Review · Reviewer_KKAN · 2024-07-20

**Soundness:** 3
**Presentation:** 3
**Contribution:** 3
**Rating:** 7
**Confidence:** 4

**Summary:**

A fundamental question in lifelong model editing of large language models (LLMs) is where the updated knowledge should reside in the model's memory. This paper identifies an inherent challenge in editing either long-term memory (direct model parameters) or working memory (non-parametric knowledge through neural network activations/representations by retrieval). It reveals that achieving reliability, generalization, and locality simultaneously in lifelong editing settings is highly challenging in existing approaches. Editing long-term memory by directly modifying parameters can lead to conflicts with irrelevant pre-trained knowledge or previous edits, compromising reliability and locality. In contrast, retrieval-based working memory edits struggle to enable the model to understand and generalize the updates effectively. To address these challenges, this paper proposes WISE, a method designed to bridge the gap between different types of memory. WISE employs a dual parametric memory scheme, comprising a main memory for pre-trained knowledge and a side memory for edited knowledge. Edits are made exclusively in the side memory, with a router trained to determine which memory to access for a given query. For continual editing, we introduce a knowledge-sharding mechanism leveraging Ties-merging, where different sets of edits are stored in distinct parameter subspaces and subsequently merged into a shared memory without conflicts.

**Strengths:**

The paper is easy to follow and nicely clarifies the challenges in lifelong model editing of LLMs in threefold: reliability, locality, and generalization, along with sufficient observations/analyses and references.

The basis for selecting later (mid-to-late) layers for side memory duplication and training sounds reasonable, and the quantitative analysis provides a reasonable demonstration of this choice.

The routing idea between long-term memory and side memory also sounds reasonable, grounded by statistical analyses.

The edit knowledge (sub)-isolation via sharding with random masks is interesting and seems to be effective in alleviating the inherent limitations that merging multiple models degenerates performance due to knowledge overlap.

In the end, the proposed method outperforms existing (lifelong) model editing baselines with improved versatility.

**Weaknesses:**

During the training phase, the model requires excessive memory since it requires copying multiple side memories with the same dimensionality as the original FFN weights in the model. Though it partially alleviates this issue by omitting copying side memory for earlier layers, it still needs substantial additional capacity.

Gradient masks for subspace memorization are randomly generated regardless of the density of target knowledge. It would be great if the model adaptively controls \rho according to the difficulty of incoming knowledge or information quantity.

**Questions:**

N/A

**Limitations:**

The paper appropriately addressed limitations and broader societal impacts.

---

> ### Author Rebuttal · Authors · 2024-08-05
>
> # Response to Reviewer KKAN
>
> Thanks for recognizing the value of our work. Your comments are highly aligned with our paper, and we hope the following comments could address your questions:
>
> > 1. Response to “During the training phase, the model requires excessive memory …”
>
> Thanks for the comment. However, it is notable that the side memory that we copy is quite small, without substantial additional capacity.
>
> - First, the memory that we copy is one linear module in one certain FFN layer, e.g., the `model.layers[34].mlp.down_proj` in **LLaMA-2-13B-chat**, so the number of parameters is quite marginal (e.g., 0.64% of the whole model).
> - Second, we introduce two variants of WISE: WISE-Merge (with only one side memory) and WISE-Retrieve (with several side memories), and both of them have limited and controllable side memories.
>     - For WISE-Merge, it only introduces marginal and *constant* additional costs (**0.64%** extra parameters and **4%** extra GPU VRAM).
>     - While WISE-Retrieve has several side memories, the experimental results show that the additional costs are also limited and controllable. Importantly, Figures 6 and 12 of our submission show that at T=3000, when there are six side memories in WISE-Retrieve, it introduces only approximately 7% inference latency and memory overhead. WISE-Retrieve also benefits from the additional memories and demonstrates higher reliability (Table 6).
>
> > 2. Response to “It would be great if the model adaptively controls $\rho$ according to the difficulty of incoming knowledge.”
>
> Your suggestion regarding adaptive control of $\rho$ based on the difficulty or information content of incoming knowledge is insightful. We briefly discuss the insights and practicality.
>
> - Recent literature [1] highlights that language models typically store approximately 2 bits of knowledge per parameter. [2] demonstrates that the performance of LoRA [3] and DoRA [4] tends to plateau and decline with an increase in trainable parameters. They underscore the importance of appropriately scaling parameters to fit each editing sample.
> - In our experiments, we observed that utilizing a mere **0.1%** of side memory parameters (about 0.04M) often resulted in unsuccessful edits per sample. Future work could explore adapting $\rho$ by identifying significant gradient values [5] or estimating the information entropy of incoming knowledge before input, aiming to enhance the robustness of WISE. We are glad to incorporate similar ideas in the future improved version of WISE.
>
> ---
>
> [1] Allen-Zhu, Zeyuan, and Yuanzhi Li. "Physics of language models: Part 3.3, knowledge capacity scaling laws." arXiv preprint arXiv:2404.05405 (2024).
>
> [2] He, Haoze, et al. "Sparse Matrix in Large Language Model Fine-tuning." arXiv preprint arXiv:2405.15525 (2024).
>
> [3] Hu, Edward J., et al. "Lora: Low-rank adaptation of large language models." ICLR 2022
>
> [4] Liu, Shih-Yang, et al. "Dora: Weight-decomposed low-rank adaptation." ICML 2024
>
> [5] Damai Dai, et al. “Knowledge Neurons in Pretrained Transformers.” ACL 2022

---

> > ### Comment · Reviewer_KKAN · 2024-08-09
> > **Thank you for your constructive rebuttal.**
> >
> > The reviewer is satisfied with the authors' detailed rebuttal. I have no remaining concerns and will keep the score.

---

> > > ### Author Response · Authors · 2024-08-12
> > > **Thanks for the post-rebuttal response**
> > >
> > > Many thanks for your time, effort, and questions! We greatly appreciate your recognition of our work and are pleased to hear that we have addressed your concerns.

---

### Author Rebuttal · Authors · 2024-08-05

# General Response

We thank all the reviewers for their time and for providing constructive comments to enhance the paper. We appreciate that reviewers recognized:

- Our paper is well-written, easy to follow, and has clear motivation (balancing Reliability, Generalization, and Locality in lifelong model editing) (Reviewers KKAN, 5h8i, fviW).
- Experimental contribution via robust and comprehensive experimental setup (several models, different datasets, and numerous baseline algorithms) (Reviewers 5h8i, pT3T).
- Solid technical contributions (reviewer fviW) and sufficient analyses (Reviewers KKAN, pT3T).

---

We include the following additional experiments in the **Rebuttal PDF of this general response**, hoping it could relieve your concerns. Specifically, the added results are as follows.

1. (**"Scale to Larger LMs (13B-chat)" experiments**) [*Reviewer pT3T*] WISE demonstrates exceptional scalability, surpassing even the performance of editing the 7B model. (Rebuttal PDF Table B)
2. (**"Ablation study on Router" experiments**) [*Reviewer pT3T*] In addition to visualizing router activations in Figure 3, we further ablate the router, demonstrating its ability to identify editing scope to ensure robust local editing. (Rebuttal PDF Table A)
3. (**"Fact Reasoning/Utilization on MQuAKE" experiments**) [*Reviewer 5h8i*] When the number of edits reaches 1000 or more, WISE (Parametric Memory) establishes superior knowledge associations, surpassing in-context editing (i.e., raw text retrieval). (Rebuttal PDF Figure A)
4. (**"Ablation study on Random Prefix Token (RPT)” experiments**) [*Reviewer fviW*] WISE economically generates random tokens from the original model, enhancing editing generalization/adaptation to diverse contexts. (Rebuttal PDF Figure B and Table C)

---

### **Common Questions.**

> 1. Response to the issue of “Ever-expanding memory” [Reviewer 5h8i] and “Lifelong model editing” [Reviewer fviW].

**Answer:**

- **The editing scope of (lifelong) knowledge editing:** As introduced in the literature, knowledge editing (i.e., model editing) serves as a knowledge injection method to bridge the gap between finetuning and timely online requirements. Also, lifelong knowledge editing (i.e., continual/continuous editing) focuses on the scenario where there are growing and sequential knowledge edits. It is worth noting that if the number of edits reaches a specific point, conducting finetuning using the accumulated data is fine, and our WISE can provide a more elegant approach by merging the side memories into the main memory by model merging techniques. In addition, in previous lifelong model editing literature, the maximal editing scope of GRACE is 3000 (T=3k) in their experiments, while in our paper, we also follow this setting.
- **Our WISE’s advantages:** Comprehensive analysis from Table 6 and Figure 11 of our original manuscript shows that WISE is capable of scaling up to more edited samples. Additionally, our method is an AI-native memory that explicitly builds knowledge connections in model parameters; therefore, it has great generalization compared with retrieval-based methods, e.g., GRACE. Also, our WISE is orthogonal to RAG (in-context editing, ICE). In Figure A of the ***Rebuttal PDF***, it is found that our method can surpass RAG/ICE and reach a higher point when combining it.
- **Remedy and future improvement:** We will give some suggestions on the memory management of WISE as a remedy for ever-expanding side memories. Akin to LRU (Least Recently Used) in Operation System, we can use similar ideas to delete or clean some side memories that are rarely accessed. In addition, merging the side memories back into the main memory by model merging techniques is also an approach; though the knowledge conflict problem may exist when doing so, the merged memory may also serve as a better initialization for finetuning the model on the editing dataset.

---

### Decision · Program_Chairs · 2024-09-25

**Decision:**

Accept (poster)

**Comment:**

This paper identifies the fundamental challenges in lifelong model editing of LLMs, where reliability, generalization and locality cannot be realized together in the lifelong editing settings. It then proposes a new method to bridge the gap between different types of memory. It develops a dual parametric memory scheme, where a main memory is for pre-trained knowledge and a side memory for edited knowledge. A router is trained to determined which memory to access for a given query. A knowledge sharing mechanism with Ties-merging is introduced for continual editing.

All the reviewers recognized that this paper is well written with comprehensive experiments. The overall proposed designs make sense and are sufficiently supported by the experiments. In addition, all the concerns raised by the reviewers are addressed in the rebuttal period. The authors are encouraged to take the feedback into account in revising the paper (e.g., making it less dense for clarity of presentation).